# Structural basis for isoform-specific kinesin-1 recognition of Y-acidic cargo adaptors

Stefano Pernigo[1†‡], Magda S Chegkazi[1†], Yan Y Yip[1], Conor Treacy[1], Giulia Glorani[1], Kjetil Hansen[2], Argyris Politis[2], Soi Bui[1], Mark P Dodding[1,3*], Roberto A Steiner[1*]

[1]Randall Centre of Cell and Molecular Biophysics, Faculty of Life Sciences and Medicine, King's College London, London, United Kingdom; [2]Department of Chemistry, King's College London, London, United Kingdom; [3]School of Biochemistry, Faculty of Life Sciences, University of Bristol, Bristol, United Kingdom

*For correspondence:
mark.dodding@bristol.ac.uk
(MPD);
roberto.steiner@kcl.ac.uk (RAS)

†These authors contributed
equally to this work

Present address: ‡Charles River
Laboratories, Saffron Walden,
United Kingdom

**Competing interests:** The authors declare that no competing interests exist.

**Abstract** The light chains (KLCs) of the heterotetrameric microtubule motor kinesin-1, that bind to cargo adaptor proteins and regulate its activity, have a capacity to recognize short peptides via their tetratricopeptide repeat domains (KLC$^{TPR}$). Here, using X-ray crystallography, we show how kinesin-1 recognizes a novel class of adaptor motifs that we call 'Y-acidic' (tyrosine flanked by acidic residues), in a KLC-isoform specific manner. Binding specificities of Y-acidic motifs (present in JIP1 and in TorsinA) to KLC1$^{TPR}$ are distinct from those utilized for the recognition of W-acidic motifs found in adaptors that are KLC- isoform non-selective. However, a partial overlap on their receptor binding sites implies that adaptors relying on Y-acidic and W-acidic motifs must act independently. We propose a model to explain why these two classes of motifs that bind to the concave surface of KLC$^{TPR}$ with similar low micromolar affinity can exhibit different capacities to promote kinesin-1 activity.

DOI: https://doi.org/10.7554/eLife.38362.001

## Introduction

Transport of cellular components along microtubules (MTs) is important in virtually all cell types and required for normal cellular activities. For neuronal function, MT-based transport is particularly critical due to the long distances that need to be travelled and strong links exist between intracellular transport and the pathogenesis of neurological diseases (*Chevalier-Larsen and Holzbaur, 2006*; *Franker and Hoogenraad, 2013*). The kinesin-1 motor, is responsible for the anterograde (plus-end directed) transport along MTs of a wide range of cargoes such as protein complexes, ribonuclear protein assemblies, vesicles, and organelles (*Hirokawa et al., 2009*). Defects in kinesin-1-mediated transport have been linked to various neurodegenerative diseases including Parkinson's, Huntington's, Alzheimer's, and hereditary spastic paraplegia (*Mandelkow and Mandelkow, 2002*; *Morfini et al., 2016*; *Morihara et al., 2014*).

At the molecular level, kinesin-1 is a tetramer consisting of two ATP-dependent motor-bearing heavy chains (KHCs) and two light chains (KLCs) that in mammalian cells are encoded by three (Kif5A-C) and four (KLC1-4) closely related genes, respectively, with distinct cell and tissue expression profiles. When not engaged in active transport, kinesin-1 is maintained in an autoinhibited state resulting from an intramolecular interaction in which the C-terminus of a single KHC tail binds at the N-terminal motor dimer interface preventing the structural transition required for ADP release (*Kaan et al., 2011*). Regulatory KLCs interact with KHCs using their N-terminal coiled-coil (CC) region that precedes an unstructured region followed by a cargo-anchoring tetratricopeptide repeat

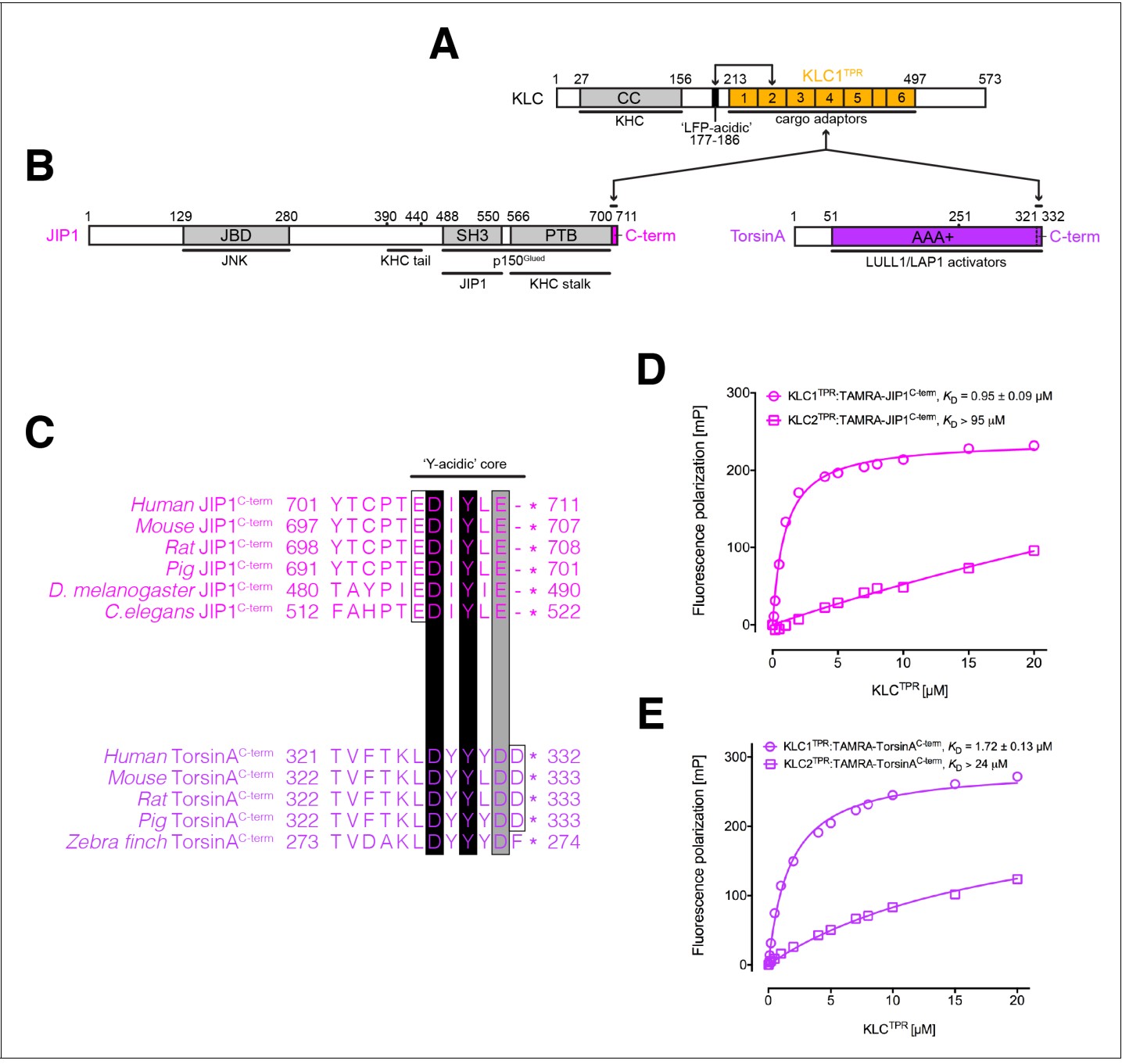

**Figure 1.** KLC1$^{TPR}$ binds Y-acidic peptides with low micromolar affinity. (**A**) Schematic diagram of KLC. Residue numbers are for the human version of the KLC1 isoform. Important domains and protein regions are highlighted as well as key interacting partners. The LFP-acidic region (in black) engages in cis with KLC$^{TPR}$ (in gold, numbers correspond to the individual TPR repeats) contributing to the auto-inhibited state. (**B**) Schematic diagram of the JIP1 cargo adaptor and TorsinA proteins. They use their respective C-terminal regions to bind to KLC1$^{TPR}$. The JIP1$^{C-term}$ (in magenta) is located immediately downstream of the PTB domain that is responsible for APP binding while TorsinA$^{C-term}$ (in violet) is an integral part of the AAA+ domain to which LULL1/LAP1 activators bind. (**C**) Multiple sequence alignment of JIP1$^{C-term}$ and TorsinA$^{C-term}$ peptides. Functionally unrelated JIP1 and TorsinA proteins share a conserved Y-acidic C-terminal region. Totally conserved Asp and Tyr residues (highlighted by black boxes), a conserved acidic residue (Glu/Asp highlighted by the grey box), and additional acidic residues (empty boxes) identify a common Y-acidic core. Residues upstream the Y-acidic core are not conserved. (**D**) Fluorescence polarization (FP) measurements indicate that TAMRA-labelled JIP1$^{C-term}$ binds with high affinity to KLC1$^{TPR}$ but not to the highly homologous KLC2$^{TPR}$. (**E**) FP measurements show that TAMRA-labelled TorsinA$^{C-term}$ behaves similarly to JIP1$^{C-term}$ with a strong preference for KLC1$^{TPR}$. The $K_D$ values reported in (**D**) and (**E**) refer to measurement carried out in buffer containing 150 mM NaCl. Error bars in the FP graphs are not visible as they are smaller than the size of the data point symbols.

DOI: https://doi.org/10.7554/eLife.38362.002

*Figure 1 continued on next page*

*Figure 1 continued*

The following source data and figure supplements are available for figure 1:

**supplement Figure 1—Source data 1.** Fluorescence polarization data at varying NaCl concentrations as presented in panels A and B.
DOI: https://doi.org/10.7554/eLife.38362.004

**Source data 1.** Fluorescence polarization data for experiments presented in panels D and E.
DOI: https://doi.org/10.7554/eLife.38362.006

**Figure supplement 1.** Binding affinity measurements between KLC1[TPR] and Y-acidic peptides at varying NaCl concentrations.
DOI: https://doi.org/10.7554/eLife.38362.003

**Figure supplement 2.** Binding of untagged JIP1[C-term] to KLC1[TPR] measured by isothermal titration calorimetry (ITC).
DOI: https://doi.org/10.7554/eLife.38362.005

domain (TPR) and a disordered C-terminal region that varies considerably between isoforms (*Figure 1A*). The unstructured region N-terminal to KLC[TPR] (TPR domain of KLC) features a leucine-phenylalanine-proline (LFP) triplet followed by Asn/Ser and flanked by negatively charged Asp/Glu residues that is highly conserved in all KLCs isoforms. We have shown that this 'LFP-acidic' region engages in cis with KLC[TPR] contributing to the maintenance of the autoinhibited state, and is likely important in cargo-driven activation (*Yip et al., 2016*).

The process of motor activation in response to cargo recognition occurs via molecular mechanisms that are incompletely understood. However, adaptor (or scaffolding) proteins that provide a molecular link between kinesin-1 and its cargoes play a clear role in coordinating motor's activity and driving transport (*Fu and Holzbaur, 2014*). While vesicular cargoes interact with adaptors that can bind to multiple sites on both KHCs and KLCs, diversity of recognition is often mediated by the interaction of the KLC[TPR] domain with short disordered peptide sequences on the adaptors. One important class of such peptides features a tryptophan residue flanked by aspartic or glutamic acid residues (for example, EWD). Experimentally validated 'W-acidic' recognition motifs are found in a growing list of adaptors including the SifA-kinesin interacting protein (SKIP), a critical host determinant in *Salmonella* pathogenesis and a regulator of lysosomal positioning, the neuronal protein calsyntenin-1 (CSTN-1), dynein intermediate chain (DIC), nesprin-2, gadkin, and cayman ataxia protein (BNIP-H) (*Aoyama et al., 2009*; *Araki et al., 2007*; *Dodding et al., 2011*; *Kawano et al., 2012*; *Konecna et al., 2006*; *Ligon et al., 2004*; *McGuire et al., 2006*; *Schmidt et al., 2009*; *Wilson and Holzbaur, 2015*). Remarkably, W-acidic motifs have an intrinsic capacity to promote kinesin-1 activity (*Dodding et al., 2011*; *Farías et al., 2015*; *Kawano et al., 2012*; *Pu et al., 2015*). We have solved the X-ray structure of KLC2[TPR] in complex with the W-acidic peptide of SKIP (SKIP[WD], sequence TNLEWDDSAI) thus deciphering the structural basis for the recognition of this important class of adaptor peptides by kinesin-1 (*Pernigo et al., 2013*).

Despite the importance of W-acidic motifs, cargo adaptors exist that do not feature this type of recognition sequence. A prominent example is the c-Jun NH$_2$-terminal kinase (JNK)-interacting protein 1 (JIP1) that is involved in the anterograde transport of the amyloid precursor protein (APP), a key determinant in Alzheimer's disease (*Matsuda et al., 2001*; *Scheinfeld et al., 2002*). The most C-terminal region of JIP1 (sequence YTCPTEDIYLE, JIP1[C-term]) has been shown to be necessary and sufficient for kinesin-1 binding and, in contrast to W-acidic sequences, JIP1[C-term] has a strong preference for the KLC1 isoform, rather than KLC2 (*Kawano et al., 2012*; *Verhey et al., 2001*; *Zhu et al., 2012*). Interestingly, JIP1[C-term] binding to KLC1[TPR] is not sufficient to promote kinesin-1 activity (*Kawano et al., 2012*). Indeed, KHC binding by JIP1 as well as the cooperation of additional proteins, like FEZ1 or JIP3, is required for cargo transport (*Blasius et al., 2007*; *Fu and Holzbaur, 2013*; *Fu and Holzbaur, 2014*; *Hammond et al., 2008*; *Satake et al., 2013*; *Sun et al., 2017*).

Using a structural approach we show here how kinesin-1 selects adaptors, like JIP1, that rely on an alternative 'Y-acidic' (tyrosine-acidic) motif for recognition. We also show how the solenoid-shaped KLC[TPR] domains utilize distinct, yet partly overlapping, portions of their concave surface to select between W-acidic and Y-acidic motifs in an isoform-specific manner following a general 'induced-fit' principle. Our work helps understanding the versatility and complexity of cargo recognition mediated by KLC[TPR] domains that depends on their unanticipated remarkable plasticity. We propose a model to explain why W-acidic and Y-acidic motifs that bind to the concave groove of KLC[TPR] with similar low micromolar affinity exhibit different capacities to promote kinesin-1 activity.

## Results

### Y-acidic C-terminal motifs bind to KLC1$^{TPR}$ with low micromolar affinity

JIP1 is an important adaptor protein that can couple kinesin-1 to several cargoes thus enabling their transport along MTs (*Figure 1B*, left-hand side). While its JNK-binding domain (JBD) is required for interaction with JNK, its PTB domain binds several proteins, including APP, apolipoprotein E receptor 2 (ApoER2), p190RhoGEF, dual leucine zipper bearing kinase (DLK), and JIP3 (JSAP1). The short JIP1$^{C-term}$ stretch that binds to KLC1$^{TPR}$ is highly conserved and characterized by a low theoretical pI value of 4.57 due to the presence of multiple acidic amino acids (*Figure 1C*, top). The most C-terminal tyrosine (Y709 in human JIP1) is particularly important for the interaction and its replacement by an alanine has been shown to result in loss of KLC1 binding in immunoprecipitation experiments (*Verhey et al., 2001*). As Y709 is embedded within the acidic patch we refer to this region as the Y-acidic core (*Figure 1C*). To further validate this interaction, we employed an in vitro fluorescence polarization (FP) assay. We first measured the affinity between TAMRA-labelled JIP1$^{C-term}$ and the highly homologous KLC1$^{TPR}$ and KLC2$^{TPR}$ domains. We find that at 150 mM NaCl TAMRA-JIP1$^{C-term}$ binds to KLC1$^{TPR}$ with a $K_D$ of 0.95 μM whilst its affinity for KLC2$^{TPR}$ is significantly lower ($K_D$ >94 μM, *Figure 1D*) consistent with a previously reported specificity of JIP1$^{C-term}$ for KLC1$^{TPR}$ (*Zhu et al., 2012*). Moreover, like we observed for W-acidic peptides (*Pernigo et al., 2013*), binding affinity measurements at varying NaCl concentrations indicate that Y-acidic motifs rely on an electrostatic component for recognition. $K_D$ values for TAMRA-JIP1$^{C-term}$:KLC1$^{TPR}$ binding are 0.48 μM and ~10 μM at NaCl concentrations of 85 and 500 mM, respectively (*Figure 1—figure supplement 1*). A control isothermal titration calorimetry (ITC) experiment carried out using unlabeled JIP1$^{C-term}$ indicates that the TAMRA label does not interfere with the interaction (*Figure 1—figure supplement 2*).

Another protein that has also been shown to bind to KLC1 in co-IP experiments is TorsinA, a constitutively inactive AAA+ (ATPases associated with a variety of cellular activities) protein whose gene is linked to early-onset dystonia type 1 (DYT1) (*Kamm et al., 2004*) (*Figure 1B*, right-hand side). As gene deletions revealed that its last ~80 amino acids are sufficient for the interaction (*Kamm et al., 2004*), we inspected the primary structure of this region and noticed that its C-terminus shares similarities with JIP1$^{C-term}$, particularly, with respect to the conservation of a Y-acidic core (*Figure 1C*, bottom). We thus hypothesised that this region might be responsible for KLC1 binding. Similarly to JIP1, we find that a TAMRA-labeled peptide encompassing the last 12 amino acids of mouse TorsinA (TAMRA-TorsinA$^{C-term}$) binds at 150 mM NaCl with $K_D$ values of 1.7 μM and >24 μM to KLC1$^{TPR}$ and KLC2$^{TPR}$, respectively (*Figure 1E*), and that affinity for KLC1$^{TPR}$ depends on the ionic strength of the medium with $K_D$ values of 1.2 μM and ~30 μM at NaCl concentrations of 85 and 500 mM, respectively (*Figure 1—figure supplement 1*). Overall, Y-acidic motifs bind to KLC1$^{TPR}$ with low micromolar affinity at physiological ionic strength and may provide a common mechanism for distinguishing between isoforms with preference for KLC1 over KLC2.

### Strategy employed for the structural elucidation of KLC1$^{TPR}$-specific Y-acidic peptide complexes

To elucidate how the well-established JIP1 cargo adaptor is recognized by KLC1$^{TPR}$ we pursued a crystallographic approach. We were previously successful in obtaining diffracting crystals of KLC2$^{TPR}$ in complex with the cognate SKIP$^{WD}$ W-acidic peptide by engineering a chimeric construct in which the peptide was fused N-terminal to KLC2$^{TPR}$ via a flexible (TGS)$_4$ linker (*Pernigo et al., 2013*). Here, we used a similar approach. However, as JIP1 binds to KLC1$^{TPR}$ using its C-terminus we hypothesized that a free carboxyl group might be important for the interaction. We therefore fused JIP1$^{C-term}$ downstream of KLC1$^{TPR}$ via a flexible (TGS)$_{10}$ linker. Crystals of the chimera grew readily in a variety of conditions. However, they exhibited very limited diffraction (worse than 15 Å resolution) or, more often, no diffraction at all. To improve crystal quality, we decided to employ Nanobodies (Nbs) as crystallization chaperones (*Pardon et al., 2014*). Using the chimera as an antigen, a pool of 29 high-affinity Nbs was generated and a subset later employed in crystallization trials following purification of each individual Nb:chimera complex. Nb-mediated crystallization dramatically improved crystal quality and allowed us to collect X-ray data for the complex to 2.7 Å resolution using synchrotron radiation. We also sought to obtain structural information on TorsinA binding. To

this end, we replaced the JIP1$^{C-term}$ sequence with that of TorsinA$^{C-term}$ and produced crystals that diffracted X-rays at 2.3 Å resolution. A summary of data collection statistics is given in *Table 1*.

## Overall structure of KLC1$^{TPR}$ in complex with Y-acidic peptides and conformational transition upon their recognition

KLC1$^{TPR}$ consists of six TPR repeats (TPR1 to TPR6), each contributed by a classical helix-turn-helix structural motif arranged in a right-handed super-helical conformation, with an additional non-TPR helix (αN) positioned between TPR5 and TPR6. The structures of the complexes reveal that the C-terminal peptides bind in an extended conformation straddling across the KLC1$^{TPR}$ concave surface with a direction opposite to that of super-helical coiling (*Figure 2*). The epitope recognized by the Nb used as crystallization chaperone is totally distinct from the peptide-binding region and maps on the outer surface with contributions from residues on TPR4(α8) and TPR5. The KLC1$^{TPR}$ domain is typically very well defined in both structures. However, most of the stretch connecting the single non-TPR αN helix to α11 of TPR6 (residues 437 – 459) could not be modeled due to flexibility. Also, electron density for helix α1 of TPR1 is of significantly poorer quality compared to all other

**Table 1.** X-ray data collection and refinement statistics.

**Data collection**

| Data set | KLC1$^{TPR}$-JIP1$^{C-term}$ | KLC1$^{TPR}$-TorsinA$^{C-term}$ |
|---|---|---|
| Beam line | P14 (Petra) | I04 (DLS) |
| Wavelength (Å) | 0.9763 | 0.9795 |
| Resolution range* (Å) | 45.18–2.70 (2.83–2.70) | 52.50–2.29 (2.83–2.70) |
| Space group | C2 | C2 |
| Cell dimensions (*a*, *b*, *c*) (Å) (α, β, γ) (°) | 107.47, 90.36, 51.70 90, 99.80, 90 | 106.04, 89.68, 50.99 90, 98.00, 90 |
| Unique reflections* | 13229 (1747) | 21275 (1021) |
| Overall redundancy* | 3.8 (3.9) | 4.7 (3.2) |
| Completeness* (%) | 98.6 (98.5) | 99.6 (93.9) |
| $R_{merge}$* (%) | 5.1 (117.6) | 8.0 (83.2) |
| $R_{p.i.m.}$ (I)* (%) | 4.7 (106.7) | 4.0 (52.0) |
| $\langle I/\sigma(I)\rangle$* | 11.9 (1.0) | 9.0 (1.1) |
| CC(1/2)* (%) | 99.9 (58.4) | 99.7 (74.1) |
| **Refinement** | | |
| PDB code | 6FUZ | 6FV0 |
| $R_{factor}$ (%) | 23.3 | 20.8 |
| $R_{free}$ (%) | 25.6 | 24.5 |
| # non-H atoms protein water heterogen | 3001 13 6 (GOL) | 3039 90 7 (PEG) |
| Average *B* value (Å$^2$) | 113.04 | 65.81 |
| rms bond lengths (Å) | 0.009 | 0.007 |
| rms bond angles (°) | 1.08 | 0.93 |

*Numbers in parentheses refer to the highest resolution bin.

DOI: https://doi.org/10.7554/eLife.38362.007

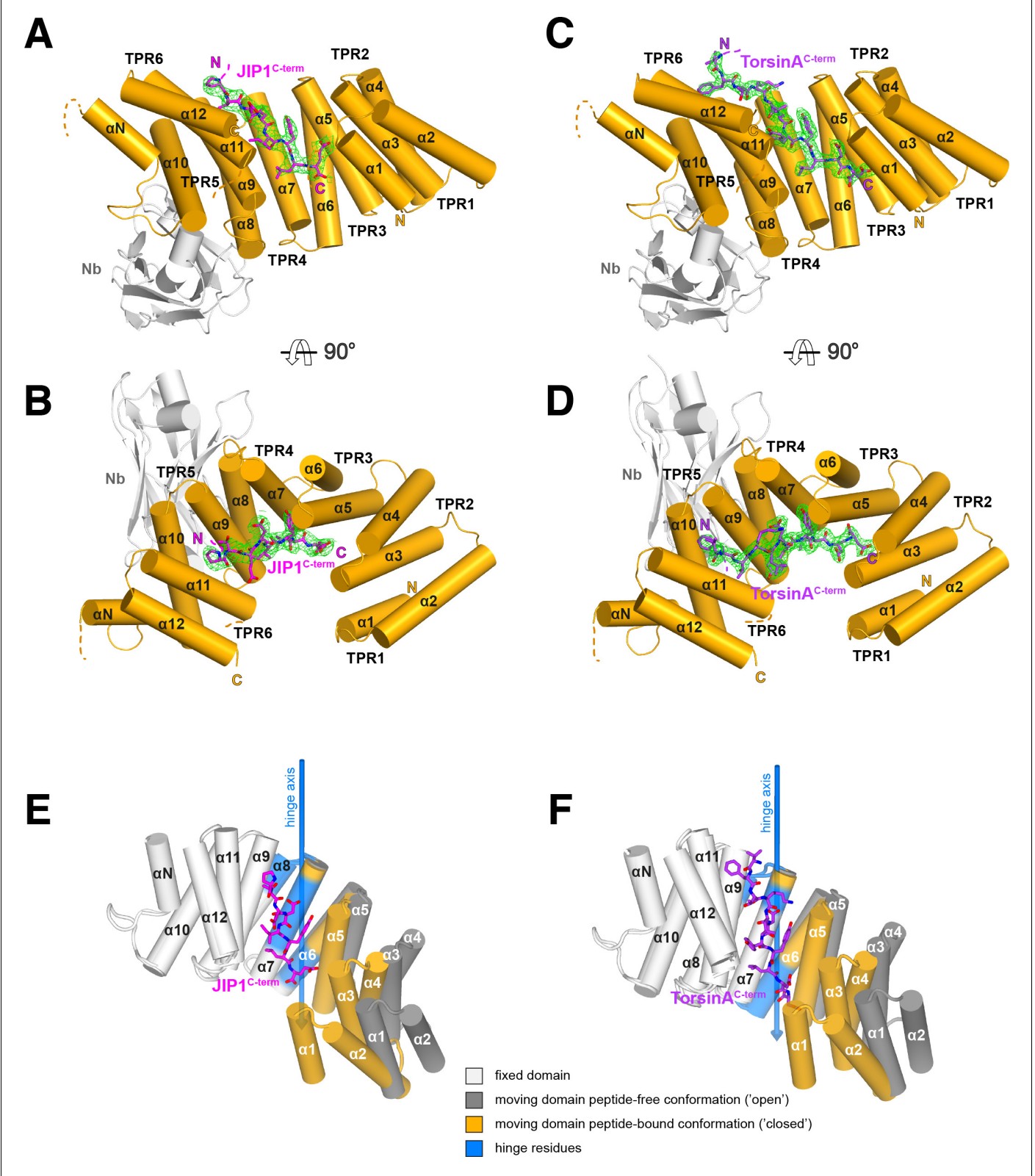

**Figure 2.** Overall structure of the complexes and conformational transition upon Y-acidic peptide recognition. (A,B) KLC1$^{TPR}$-JIP1$^{C-term}$. Illustrated representations of the JIP1$^{C-term}$ peptide (displayed as a magenta stick model accompanied by its 2$mF_o$-$DF_c$ electron density in green contoured at the 1.1σ level) bound to the KLC1$^{TPR}$ domain (gold) in two orthogonal views. The Nanobody (Nb) is shown in light grey. Helices α1 to α12 of the six TPR helix-turn-helix motifs and the non-TPR helix αN located between TPR5 and TPR6 are labeled. The flexible region between αN and α12 is indicated

*Figure 2 continued on next page*

*Figure 2 continued*

with a broken line while the unmodeled engineered linker connecting α12 to the JIP1[C-term] is not shown for clarity. Color-coded N and C labels indicate the N- and C-termini, respectively. Noncarbon elements are nitrogen and oxygen in dark blue and red, respectively. (C,D) KLC1[TPR]-TorsinA[C-term]. As (A, B) for the TorsinA[C-term] peptide colored in violet. (E,F) Conformational transition in the KLC1[TPR] domain upon recognition of Y-acidic peptides. Superposition of ligand-free KLC1[TPR] (PDB code 3NF1) and KLC1[TPR]-JIP1[C-term] (E) or KLC1[TPR]-TorsinA[C-term] (F) performed using the DynDom algorithm (*Poornam et al., 2009*). The analysis reveals the presence of an N-terminal 'moving' domain (highlighted in dark grey and gold for KLC1[TPR] and the KLC1[TPR]-Y-acidic complexes, respectively) and a C-terminal fixed domain (in light grey) with 'bending residues' shown in blue. The conformational transition requires a rotation of ~30° degrees around the hinge axis represented by a blue arrow. Hinge residues are also highlighted in blue.
DOI: https://doi.org/10.7554/eLife.38362.008

The following figure supplement is available for figure 2:

**Figure supplement 1.** $2mF_o$-$DF_c$ electron density of selected TPR helices in the KLC1[TPR]-TorsinA[C-term] complex.
DOI: https://doi.org/10.7554/eLife.38362.009

TPR helices indicating that this region is more disordered (*Figure 2—figure supplement 1*). Indeed, this helix appears to display an unusual degree of plasticity (*Nguyen et al., 2017*). For JIP1[C-term], electron density is observed for the last eight residues of the 11-aa long peptide (*Figure 2A,B*) whilst for TorsinA[C-term] we could model all but the first residue of the 12-aa long peptide (*Figure 2C,D*). The engineered (TGS)$_{10}$ linker connecting the TPR domain to the Y-acidic cargo adaptor peptides is not visible in electron density maps suggesting that this region is flexible and does not interfere with binding.

A comparison between the structures of KLC1[TPR] in the unbound (*Zhu et al., 2012*) and Y-acidic peptide-loaded states reveals dramatic conformational differences whereby peptide recognition induces an 'open'-to-'closed' transition in its solenoid structure (*Figure 2E,F*) (*Figure 2-Video 1*). While the KLC1[TPR] domains are essentially identical in both Y-acidic peptide-bound structures (rmsd = 0.75 Å over 259 equivalent Cα atoms), a similar global superposition between the peptide-free ('open', PDB code 3NF1) and the TorsinA[C-term]-bound ('closed') structures gives a rmsd of ~4.1 Å for 253 equivalent Cα atoms. An analysis using the DynDom algorithm (*Poornam et al., 2009*) indicates that peptide recognition elicits a conformational transition that is best described by a ~30° rotation of an N-terminal domain with respect to a hinge region largely contributed by residues belonging to second-half and first-half of TPR3(α6) and TPR4(α8) (residues 323 – 329, 348 – 350, 353 – 358), respectively, while the C-terminal region (second-half of α8 to α12) is fixed. As a result, the TPR1(α1) helix axis undergoes a lateral displaced of over 16 Å. Each individual domain behaves essentially as a rigid unit (rmsd N-terminal moving domains = 1.06 Å over 117 equivalent Cα atoms, rmsd C-terminal fixed domains = 0.88 Å over 131 equivalent Cα atoms). Overall, this highlights a significant degree of plasticity of the KLC1[TPR] domain that is used to accommodate the Y-acidic peptides by an 'induced fit' mechanism.

## KLC1[TPR]-Y-acidic peptide interface

JIP1[C-term] and TorsinA[C-term] peptides are recognized by their KLC1[TPR] receptor using a common binding mode. An analysis performed with the Protein Interfaces, Surfaces and Assemblies (PISA) algorithm (*Krissinel and Henrick, 2007*) indicates that ~75% of the surface atoms of the bound peptides form an extensive interface that is stabilized by residues contributed by the first helices of TPR2-6 (α3, α5, α7, α9, α11) whilst fewer residues on α4, α6, α8, α10 make secondary contacts (*Figure 3A*). Overall, the interface area is ~710 Å² and ~887 Å² for the KLC1[TPR]-JIP1[C-term] and KLC1[TPR]-TorsinA[C-term] complexes, respectively. The larger surface area for the latter complex reflects the longer TorsinA peptide bound to KLC1[TPR]. As JIP1 is one of the most important and extensively validated cargo adaptors we will discuss general principles for Y-acidic peptide binding using the KLC1[TPR]-JIP1[C-term] complex as a reference.

The structure shows that the recognition peptide binds in an extended conformation to the positively charged inner surface of the TPR solenoid structure (*Figure 3B*) with the conserved tyrosine within the Y-acidic core (Y709, referred to as position 0, $p^0$) hosted by a 'made-to-measure' cavity (highlighted by a white dotted line in

**Video 1.** KLC1[TPR] conformational transition upon TorsinA[C-term] binding.
DOI: https://doi.org/10.7554/eLife.38362.010

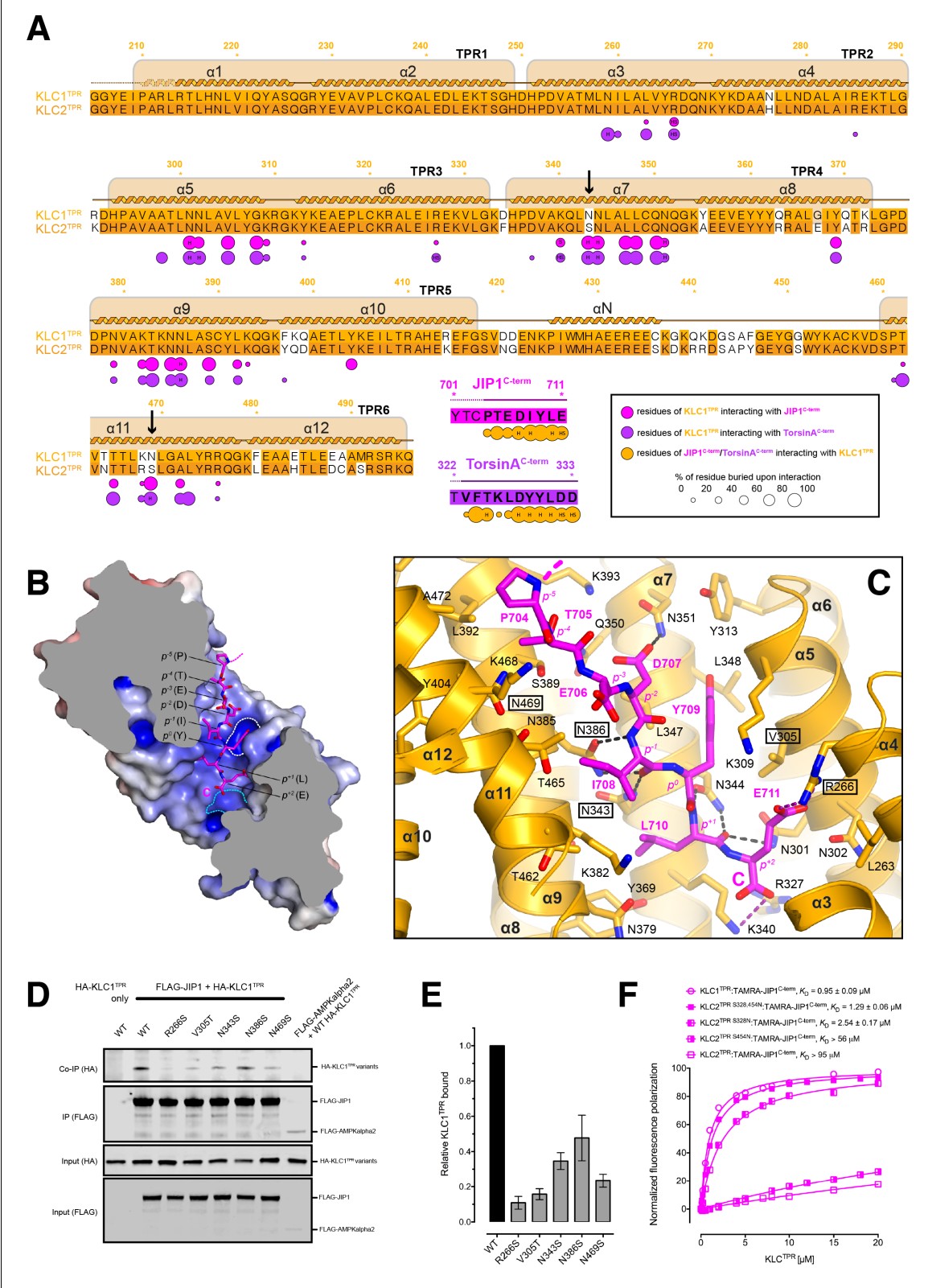

**Figure 3.** Molecular interface and cellular validation. (**A**) Interface contact map for the KLC1^TPR-JIP1^C-term/TorsinA^C-term complexes. The sequence of KLC2^TPR is also shown for comparison. Colored circles indicate residues at the interface as indicated in the inset. Their radius is proportional to the buried area. The letter codes H and S highlight residues involved in hydrogen bonds and salt bridges, respectively. The black arrows highlight the only two residues making significant contacts with the Y-acidic peptides that are not conserved in KLC2^TPR. Numbers are for KLC1^TPR. (**B**) Sliced-surface view

Figure 3 continued

of the KLC1$^{TPR}$ domain with bound JIP1$^{C-term}$. The molecular surface is colored according to its electrostatic potential. Positive and negative potential in the +10$_k$BT/e to −10$k_B$T/e range is shown in blue and red, respectively. (C) Details of the KLC1$^{TPR}$-JIP1$^{C-term}$interface. The amino acids involved in the interactions are shown as stick models with oxygen atoms colored in red, nitrogen in blue and carbon in gold and magenta for KLC1$^{TPR}$ and JIP1$^{C-term}$, respectively. Hydrogen bonds and salt bridges are represented by dotted grey and magenta lines, respectively. Residues tested for cellular validation (see panel D) are boxed. (D) Western blot analysis of coimmunoprecipitation assays showing the effect of selected KLC1$^{TPR}$ mutations on the interaction with JIP1. (E) Quantification of relative binding from two independent coimmunoprecipitation experiments. Error bars show SEM. All selected mutations reduce binding between ~65% and 90% compared to wild-type (WT) and are statistically significant to p<0.001. Statistical analysis was performed with the GraphPad Prism package using ordinary one-way ANOVA for multiple comparisons. (F) FP measurements show that the KLC2$^{TPR\ S328,454N}$ variant binds JIP1$^{C-term}$ with an affinity comparable to KLC1$^{TPR}$ with the S328N substitution playing a dominant role in restoring KLC1$^{TPR}$-like affinity. Residues S328 and S454 of KLC2 correspond to residues N343 and N469 of KLC1$^{TPR}$, respectively. Binding curves for KLC1$^{TPR}$ and KLC2$^{TPR}$ are also shown for comparison.

DOI: https://doi.org/10.7554/eLife.38362.011

The following source data and figure supplements are available for figure 3:

**supplement Figure 1—Source data 1.** Fluorescence polarization data for TAMRA-JIP1 $^{amidated\ C-term}$ binding to KLC1$^{TPR}$.

DOI: https://doi.org/10.7554/eLife.38362.013

**Source data 1.** Quantification of relative KLC1$^{TPR}$ binding in three independent coimmunoprecipitation experiments as shown in panel E.

DOI: https://doi.org/10.7554/eLife.38362.019

**Source data 2.** Normalized fluorescence polarization and SEM for experiments presented in panel F.

DOI: https://doi.org/10.7554/eLife.38362.020

**Figure supplement 1.** C-terminal amidation of JIP1$^{C-term}$ abrogates KLC1$^{TPR}$ binding.

DOI: https://doi.org/10.7554/eLife.38362.012

**Figure supplement 2.** KLC1$^{TPR}$-TorsinA$^{C-term}$ interface.

DOI: https://doi.org/10.7554/eLife.38362.014

**Figure supplement 3.** Superposition of KLC1$^{TPR}$-bound JIP1$^{C-term}$ and TorsinA$^{C-term}$ peptides.

DOI: https://doi.org/10.7554/eLife.38362.015

**Figure supplement 4.** Control coimmunoprecipitation assays.

DOI: https://doi.org/10.7554/eLife.38362.016

**Figure supplement 5.** Fluorescence polarization measurements for the binding of TAMRA-TorsinA$^{C-term}$ to the KLC2$^{TPR\ S328,454N}$ variant.

DOI: https://doi.org/10.7554/eLife.38362.017

**Figure supplement 5—source data 1.** Fluorescence polarization data for TAMRA-TorsinA$^{C-term}$ binding to KLC2$^{TPR\ S328,454N}$ .

DOI: https://doi.org/10.7554/eLife.38362.018

*Figure 3B*). The cavity stabilizing the side-chain of ($p^0$)Y709, located between TPR3(α5/α6) and TPR4 (α7), is lined by V305, K309, Y313, N344, L347, L348, N351 with the amide group of the latter residue participating in a H-bond with ($p^{−2}$)D707(Oδ2) (*Figure 3C*). Two asparagine pairs (N386/N343 and N301/N344) also engage in hydrogen bonds the main chain atoms of ($p^{−1}$)I708 and ($p^{+1}$)L710, respectively, while the C-terminal glutamate (E711) at position ($p^{+2}$) is locked in place by salt bridges with residues R266 and K340 that interact with its side-chain and main-chain carboxylate groups, respectively. The latter interaction is critically important as FP measurement using a C-terminally amidated version of the TAMRA-JIP1$^{C-term}$ peptide (TAMRA-JIP1$^{amidated\ C-term}$) shows complete abrogation of KLC1$^{TPR}$ binding (*Figure 3—figure supplement 1*). Virtually all key interactions observed in the KLC1$^{TPR}$-JIP1$^{C-term}$ structure also stabilize the bound TorsinA$^{C-term}$ peptide (*Figure 3—figure supplement 2*). The additional amino acid at the $p^{+3}$ position present in TorsinA$^{C-term}$ is hosted by a cavity (highlighted by a cyan dotted line in *Figure 3B*) lined partly by yet another asparagine pair (N259/N302) that together with the side-chains of R327 and K340 stabilize the C-terminal aspartate residue (*Figure 3—figure supplement 2*). As TorsinA$^{C-term}$ binds to KLC1$^{TPR}$ with comparable affinity this indicates that the loss of the free carboxylate at position $p^{+2}$ is largely compensated by the gain of the additional C-terminal acidic residue at $p^{+3}$. While JIP1$^{C-term}$ and TorsinA$^{C-term}$ peptides display a virtually identical main-chain trace for their Y-acidic core, conformational variability is observed for their N-terminal portion that is not conserved (*Figure 3—figure supplement 3*). Overall, a combination of electrostatics, specific interactions often mediated by receptor asparagine residues, and the intrinsically malleable nature of the cargo adaptor peptides, plays a main role in the recognition process.

## Cellular validation of the KLC1$^{TPR}$-JIP1$^{C\text{-}term}$ interface and molecular determinants of KLC1 specificity

To further validate the KLC1$^{TPR}$-JIP1$^{C\text{-}term}$ interface and to rationalize the strong preference of this cargo adaptor for KLC1$^{TPR}$ over the highly homologous KLC2$^{TPR}$ observed in FP measurements, we employed immunoprecipitation in HeLa cells in combination with structure-guided mutagenesis. Cells were co-transfected with wild-type and mutant constructs expressing hemagglutinin (HA)– KLC1$^{TPR}$ and FLAG-JIP1 (*Figure 3D*). For mutagenesis, key KLC1$^{TPR}$ interface residues R266, N343, N386, N469 were substituted with a serine to alter their salt bridge/H-bonding capabilities while V305 was replaced with a threonine to generate an isosteric change with altered polarity. We also mutagenized residues not at the interface as negative controls (Q222A, E484A, E488A) Overall, we find that all amino acid substitutions at the interface, but not negative controls (*Figure 3—figure supplement 4*), substantially reduce the affinity between KLC1$^{TPR}$ and JIP1 with residual binding quantified ranging from ~15% to ~35% compared to wild-type (*Figure 3E*). In particular, a very strong effect is observed for the R266S replacement that prevents the formation of a salt bridge with the (p$^{+2}$) acidic side-chain and a similar near-abrogation of binding is engendered by an increase in polarity (V305T) within the receptor cavity surface that interacts 'face-on' with the aromatic ring of (p$^0$) tyrosine (*Figure 3C*). The N343S and N469S substitutions are particularly informative with respect to the specificity of Y-acidic peptides for the KLC1 isoform over KLC2. A sequence comparison between KLC1$^{TPR}$ and KLC2$^{TPR}$ highlights that the only two interface residues that differ between these isoform domains are KLC1 N343 and N469. These are naturally present as serine residues (S328 and S454, respectively) in KLC2 (highlighted by arrows in *Figure 3A*). The structure shows that asparagines at these topological positions play a key role in maintaining the stability of the complex via H-bond interactions at positions $p^{-5}$ and $p^{-1}$ of the main chain cargo peptide. Our co-IP experiment shows that a substitution at either of these positions by a serine, thus imparting KLC2$^{TPR}$-like properties, negatively affects the stability of the complex. To further probe the role of these residues in KLC1 specificity, we have generated KLC2$^{TPR\ S328N}$, KLC2$^{TPR\ S454N}$, and KLC2$^{TPR\ S328,454N}$ variants and used them in FP measurements to determine their affinity for TAMRA-JIP1$^{C\text{-}term}$ (*Figure 3F*). We find that both individual substitutions increase the affinity for the JIP1$^{C\text{-}term}$ peptide with S328N playing a main role (~37 fold increase in activity compared to KLC2$^{TPR}$). Importantly, the KLC2$^{TPR\ S328,454N}$ double variant binds TAMRA-JIP1$^{C\text{-}term}$ with an affinity comparable to KLC1$^{TPR}$ ($K_D$ = 1.29 µM) and the same is true for TAMRA-TorsinA$^{C\text{-}term}$ binding (*Figure 3—figure supplement 5*). Therefore, residues N343 and N469 are entirely responsible for the specificity of Y-acidic motif for KLC1.

## Y-acidic and W-acidic motifs bind at partly overlapping sites on KLC$^{TPR}$ and are differentially affected by the presence of the LFP-acidic intramolecular region

We have previously solved the X-ray structure of KLC2$^{TPR}$ in complex with the SKIP$^{WD}$ W-acidic peptide, thus providing a structural basis for the recognition of this widespread adaptor motif (*Pernigo et al., 2013*). A comparative analysis with the structures presented here reveals that Y-acidic and W-acidic motifs are recognized by partly overlapping binding sites that are shifted with respect to each other along the inner channel generated by the TPR right-handed super-helical arrangement (*Figure 4A*). The W-acidic motif binds toward the N-terminal end of the channel with its critical tryptophan residue sandwiched between TPR2 and TPR3 while the Y-acidic motif is positioned more C-terminally with its key tyrosine residue stabilized by the interface between TPR3 and TPR4 approximately at the same location occupied by the SKIP$^{WD}$ leucine at (p$^{-2}$). Although W-acidic binding also occurs by an 'induced fit' mechanism, it elicits a more restrained closure of the receptor compared to Y-acidic binding, defined by smaller rotation (~20.0°) with respect to an alternative, more N-terminal, hinge axis (*Figure 4—figure supplement 1*). Despite these differences, W-acidic and Y-acidic binding sites partly overlap and in agreement with this we find that an unlabeled SKIP$^{WD}$ peptide is able to specifically compete TAMRA-JIP1$^{C\text{-}term}$ binding in a concentration-dependent manner in our FP assay (*Figure 4B*). The functional implication of this is that a single KLC1$^{TPR}$ is not expected to simultaneously engage both types of cargo adaptors. The partial separation of Y-acidic and W-acidic binding sites on KLC$^{TPR}$ has also a possible important functional consequence. We have solved previously the 4 Å-resolution structure of KLC2$^{extTPR}$, an extended TPR

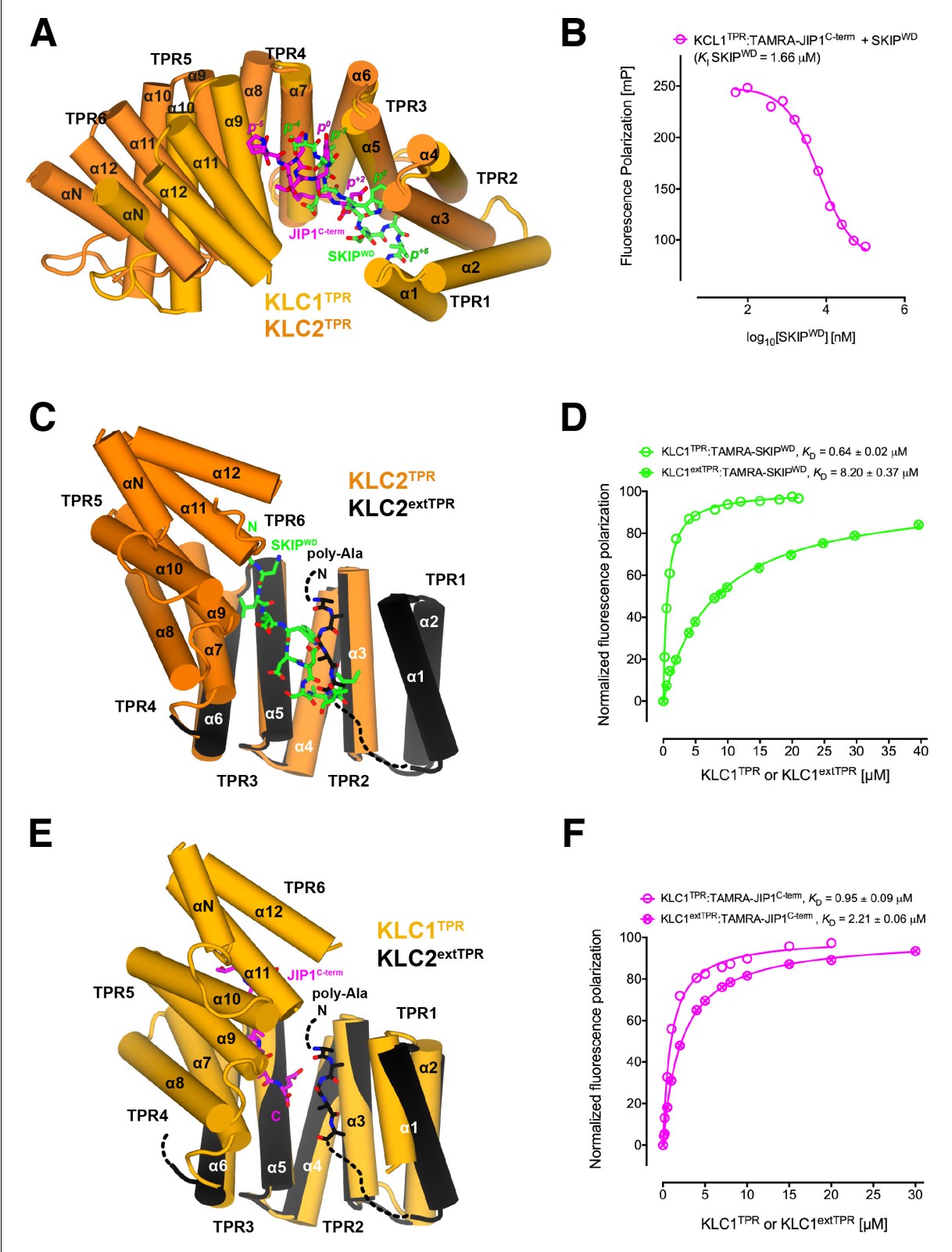

**Figure 4.** Y-acidic and W-acidic motifs bind at partly overlapping sites on KLC$^{TPR}$ and are differentially affected by the presence of the LFP-acidic intramolecular region. (**A**) Superposition of the KLC1$^{TPR}$-JIP1$^{C-term}$ and KLC2$^{TPR}$-SKIP$^{WD}$ (*Pernigo et al., 2013*) complexes. JIP1$^{C-term}$ and SKIP$^{WD}$ are shown as stick representation and colored in magenta and green, respectively. Selected amino acid positions are labeled using the Y-acidic tyrosine and the W-acidic tryptophan as references (position 0, $p^0$). KLC1$^{TPR}$ and KLC2$^{TPR}$ are colored in gold and orange, respectively. TPR1 is missing in the

*Figure 4 continued on next page*

*Figure 4 continued*

KLC2$^{TPR}$-SKIP$^{WD}$ structure. The positions of the Y-acidic JIP1$^{C-term}$ and W-acidic SKIP$^{WD}$ peptides are shifted along the inner cavity of the super-helical KLC$^{TPR}$ architecture and binding at these alternative locations induces a different degree of conformational change in the receptor. (B) Titration of increasing amounts of unlabeled SKIP$^{WD}$ in the KLC1$^{TPR}$:JIP1$^{C-term}$ complex shows a decreased FP signal that supports competitive displacement owing to their overlapping binding sites. (C) Superposition of the KLC2$^{TPR}$-SKIP$^{WD}$ (*Pernigo et al., 2013*) and KLC2$^{extTPR}$ (*Yip et al., 2016*) structures (view is rotated by ~90° around the *x* axis compared to A). For clarity, only the first three TPRs of KLC2$^{extTPR}$ (in black) that define the local environment of the LFP-acidic (also in black, modelled as poly-Ala) are shown. SKIP$^{WD}$ and LFP-acidic peptides bind at overlapping positions. (D) FP measurements show that the N-terminal extension of KLC1$^{TPR}$ domain that includes the LFP-acidic region (KLC1$^{extTPR}$) inhibits the interaction with TAMRA-SKIP$^{WD}$ compared with KLC1$^{TPR}$ alone. (E) Superposition of the KLC1$^{TPR}$-JIP1$^{C-term}$ and KLC2$^{extTPR}$ (*Yip et al., 2016*) structures (same orientation as C). As in (C) only the first three TPRs of KLC2$^{extTPR}$ are shown in black. JIP1$^{C-term}$ (in magenta) and LFP-acidic (in black) peptides bind at largely independent positions. (F) FP measurements show that the LFP-acidic N-terminal extension to KLC1$^{TPR}$ has only a modest impact on TAMRA-JIP1$^{C-term}$ binding (*Yip et al., 2016*). This panel is re-produced under the following license agreement: http://www.pnas.org/sites/default/files/advanced-pages/authorlicense.pdf. It is not available under CC-BY and is exempt from the CC-BY 4.0 license.

DOI: https://doi.org/10.7554/eLife.38362.021

The following source data and figure supplement are available for figure 4:

**Source data 1.** Fluorescence polarization data for the competition of TAMRA-JIP1$^{C-term}$ with increasing amounts of unlabeled SKIP$^{WD}$ as shown in panel B.

DOI: https://doi.org/10.7554/eLife.38362.023

**Source data 2.** Normalized fluorescence polarization data and SEM for the experiments presented in panels E and F.

DOI: https://doi.org/10.7554/eLife.38362.024

**Figure supplement 1.** Conformational change in the KLC2$^{TPR}$ domain upon SKIP$^{WD}$ W-acidic peptide binding.

DOI: https://doi.org/10.7554/eLife.38362.022

featuring the conserved LFP-acidic region N-terminal to the KLC$^{TPR}$ domain that contributes to kinesin-1 inhibition (*Figure 1A*) (*Yip et al., 2016*). In this structure the KLC2$^{TPR}$ domain is in its 'open' (ligand-free) conformation and we could model a stretch five poly-Ala residues between helices α1, α3, α5 that likely defines the general location of the intramolecular LFP-acidic region. The poly-Ala peptide partly overlaps with the SKIP$^{WD}$ binding site (*Figure 4C*) and consistent with this we found that a TAMRA-SKIP$^{WD}$ peptide binds in an FP assay with lower affinity to KLC2$^{extTPR}$ compared to KLC2$^{TPR}$ due to the inhibitory effect of the intramolecular LFP-acidic stretch (*Yip et al., 2016*). We have now also measured the affinity of TAMRA-SKIP$^{WD}$ for KLC1$^{extTPR}$ and KLC1$^{TPR}$ and, like for KLC2, the presence of the LFP-acidic region significantly inhibits binding, in this case, by more than one order of magnitude ($K_D$ from $0.64 \pm 0.02$ µM to $8.20 \pm 0.37$ µM, *Figure 4D*). In contrast, TAMRA-JIP1$^{C-term}$ binds with a similar affinity to either KLC1$^{extTPR}$ ($K_D = 2.21 \pm 0.06$ µM) or KLC1$^{TPR}$ ($K_D = 0.95 \pm 0.09$ µM) (*Figure 4F*) (*Yip et al., 2016*). A structural superposition between KLC1$^{TPR}$-JIP1$^{C-term}$ and KLC2$^{extTPR}$ allows to rationalize the minimal impact that the LFP-acidic extension has on JIP1$^{C-term}$ binding. We observe that the KLC1$^{TPR}$ N-terminal domain rotation elicited by JIP1$^{C-term}$ binding is compatible with the presence of the poly-Ala peptide as no clashes are observed (*Figure 4E*). This is largely due to the more advanced position along the inner KLC$^{TPR}$ cavity of this adaptor peptide compared to SKIP$^{WD}$. Thus, although with the obvious limitation imposed by a poly-Ala model that does not allow to reveal the full range of possible intermolecular interactions, we suggest that while an intramolecular interaction with the LFP-acidic region is not compatible with SKIP$^{WD}$ binding, this might be of little or no interference with JIP1$^{C-term}$ loading. This is in agreement with FP data (*Figure 4F*).

## JIP1$^{C-term}$ loading is important for JIP3 recruitment but JIP1/JIP3 co-operativity requires additional interactions outside the KLC1$^{TPR}$ domain

In mammals, the JIP family of adaptor proteins is composed of four members with JIP1 and JIP2 sharing a similar domain organization that is different from that of JIP3/JIP4 (*Whitmarsh, 2006*). At their N-terminal region, both JIP3/4 are composed of leucine zipper segments, the second of which (JIP3$^{LZ2}$), has also been found to mediate interaction with KLC1$^{TPR}$ (*Bowman et al., 2000*; *Kelkar et al., 2005*; *Nguyen et al., 2005*). Functionally, various studies have shown that JIP1 and JIP3 cooperate for kinesin-1-mediated transport (*Hammond et al., 2008*; *Satake et al., 2013*; *Sun et al., 2017*). To investigate this cooperative effect in the context of our study, we performed co-immunoprecipitation studies (*Figure 5A*). We find that while JIP1 interacts effectively with

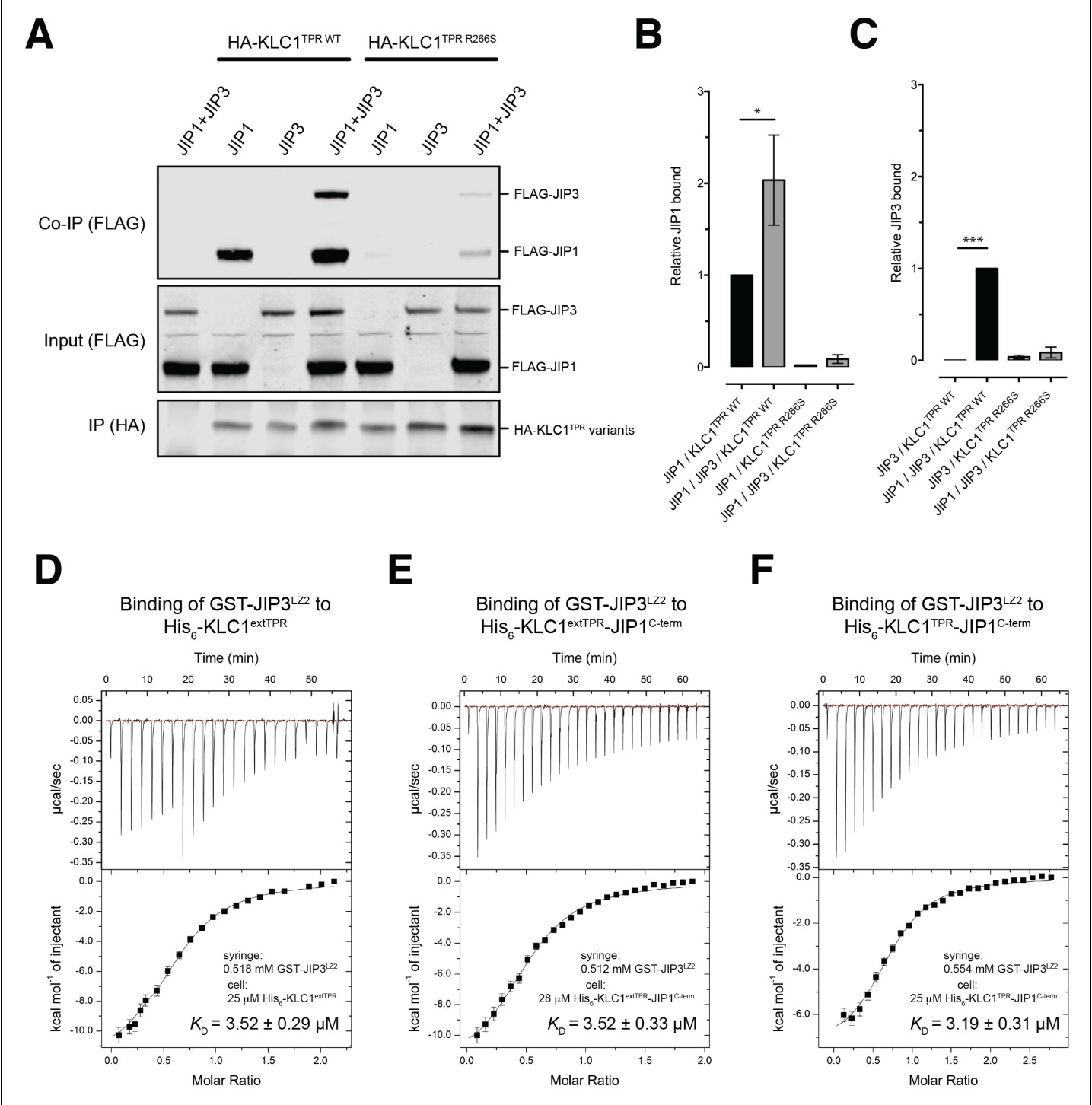

**Figure 5.** JIP1/JIP3 co-operativity. (**A**) Western blot analysis of coimmunoprecipitation assays showing co-operative formation of the JIP1 and JIP3 complex. The effect of the KLC1$^{TPR\ R266S}$ mutations on the formation of the complex is also shown. (**B, C**) Quantification of relative JIP1 binding (**B**) and JIP3 binding (**C**) from three independent co-immunoprecipitation experiments (normalized to the sample indicated by a black bar). Error bars show SEM. Statistical analysis was performed with the GraphPad Prism package using ordinary one-way ANOVA for multiple comparisons, *$p < 0.05$, ***$p < 0.001$. (**D, E, F**) Binding of GST-JIP3$^{LZ2}$ to His$_6$-KLC1$^{extTPR}$ (**D**), His$_6$-KLC1$^{extTPR}$-JIP1$^{C-term}$ (**E**), His$_6$-KLC1$^{TPR}$-JIP1$^{C-term}$ (**F**) measured by ITC. It shows that JIP3$^{LZ2}$ affinity for the TPR domain is not affected by either the LFP-acidic region or JIP1$^{C-term}$ loading. All experiments were carried out in 50 mM HEPES, 500 mM NaCl, 5 mM 2-mercaptoethanol at T = 20°C. In (**D**) the volume of the first six injections following the initial sacrificial one was 1.0 µl and the rest were 2.0 µl. In (**E**) and (**F**) all working injections were 1.5 µl. All injections were performed at 150 s time intervals.

DOI: https://doi.org/10.7554/eLife.38362.025

*Figure 5 continued on next page*

*Figure 5 continued*

The following source data and figure supplement are available for figure 5:

**Source data 1.** Quantification of relative JIP1 and JIP3 binding in three independent coimmunoprecipitation experiments as shown in panels B and C, respectively.
DOI: https://doi.org/10.7554/eLife.38362.027

**Figure supplement 1.** Native mass spectra of the untagged SEC-purified KLC1$^{TPR}$-JIP1$^{C-term}$:JIP3$^{LZ2}$ complex.
DOI: https://doi.org/10.7554/eLife.38362.026

KLC1$^{TPR}$ in isolation this is not the case for JIP3, suggesting that the JIP3$^{LZ2}$-KLC1$^{TPR}$ interaction is weaker. However, co-expression of both adaptor proteins significantly enhances their association in keeping with a co-operative behavior. Moreover, JIP1$^{C-term}$ anchoring to KLC1$^{TPR}$ is critically important for the assembly of the ternary complex as inhibition of JIP1 binding to KLC1$^{TPR}$ by the R266S mutation, significantly reduced JIP3 association. These results are quantified in *Figure 5B,C*.

As JIP1$^{C-term}$ binding alters the conformation of KLC1$^{TPR}$, we sought to explore whether this affects binding of JIP3$^{LZ2}$ to KLC1$^{TPR}$ using ITC measurements (*Table 2*). Binding of GST-JIP3$^{LZ2}$ to His$_6$-KLC1$^{extTPR}$ is an enthalpy-driven process with a dissociation constant $K_D$ of ~3.5 µM in which two TPR domains bind to a JIP3$^{LZ2}$ coiled-coil dimer (*Figure 5D*). This stoichiometry is consistent with the recently published crystallographic analysis of the KLC2$^{TPR}$-JIP3$^{LZ2}$ complex that shows two TPR domains symmetrically bound to the JIP3$^{LZ2}$ coiled-coil (*Cockburn et al., 2018*) and also native mass spectrometry (MS) analysis (*Figure 5—figure supplement 1*). No significant changes in $K_D$ are observed when His$_6$-KLC1$^{extTPR}$ is replaced by either His$_6$-KLC1$^{extTPR}$-JIP1$^{C-term}$ (*Figure 5E*) or His$_6$-KLC1$^{TPR}$-JIP1$^{C-term}$ (*Figure 5F*) thus indicating that the binding of JIP3$^{LZ2}$ to KLC1$^{TPR}$ is independent of either JIP1$^{C-term}$ and of the conformational change it elicits or the LFP-acidic region. Taken together, these data support the notion that JIP1 and JIP3 act co-operatively to recruit kinesin-1 and that this requires KLC1$^{TPR}$ binding to JIP1$^{C-term}$. However, key interactions mediating this effect lie outside of the direct KLC$^{TPR}$ domain interfaces.

## Discussion

For many cargo types, anchoring to the kinesin-1 motor is not direct but mediated by adaptor or scaffolding proteins that provide a molecular and mechanistic link for kinesin-driven intracellular transport (*Fu and Holzbaur, 2014*). An important region on the regulatory light chains that is often engaged by adaptors is their KLC$^{TPR}$ domain. TPR domains are found in a wide range of proteins and represent common platforms for protein-protein interactions (D'Andrea and Regan, 2003; *Zeytuni and Zarivach, 2012*). Like some other repeat-containing proteins, such as leucine-rich-repeats and WD40s, TPR domains have been traditionally considered fairly rigid docking platforms that respond with limited conformational alterations to ligand binding, thus lending support to the view that they are essentially pre-organized architectures onto which specific ligand-binding residues are 'grafted' (*Cortajarena and Regan, 2006*). Such a static view has been more recently mitigated by examples of TPR domains that undergo some conformational changes in response to protein-protein interactions. For example, a reduction of the solenoid super-helical pitch has been observed in the TPR domain of the Pex5p receptor in response to loading the peroxisomal targeting signal type 1 (PTS1) (*Stanley et al., 2006*). Also, in the magnetosome-associated protein, MamA the N-terminal portion of its TPR domain undergoes a radial movement of ~3° upon binding of a putative ligand imitator (*Zeytuni et al., 2011*). Our crystallographic analysis of the recognition mechanism of Y-acidic motifs by KLC1$^{TPR}$ that extends our previous work on the recognition of W-acidic motifs (*Pernigo et al., 2013*), strongly challenges the view of TPR motifs as rigid scaffolds. Instead, it portrays their strongly adaptive nature as binding of these motifs involves rotations of the N-terminal portion of the molecule of up to ~30° with respect to alternative hinge axes. Overall, KLC$^{TPR}$ domains display the remarkable ability to employ inter-TPR repeats as separate 'compartments' to stabilize different peptide types in an isoform-specific manner.

While JIP1 is one of the best-established and most studied adaptors, the role for TorsinA in the context of kinesin-1-based transport is less clear. TorsinA, the founding member of the Torsin family, is a constitutively inactive AAA+ protein of medical importance as a three-base pair (ΔGAG) deletion that removes one of a pair of glutamic acid residues (Glu-302/303) in proximity of its C-terminus is

**Table 2.** ITC parameters.

| GST-JIP3$^{LZ2}$ interactions with | N | $K_D$ (μM) | ΔH (kcal/mol) | TΔS (kcal/mol) | ΔG (kcal/mol) |
|---|---|---|---|---|---|
| His$_6$-KLC1$^{extTPR}$ | 0.59 ± 0.01 | 3.52 ± 0.29 | −13.19 ± 0.38 | −5.86 | −7.33 |
| His$_6$-KLC1$^{extTPR}$-JIP1$^{C-term}$ | 0.52 ± 0.01 | 3.52 ± 0.33 | −12.69 ± 0.45 | −5.39 | −7.30 |
| His$_6$-KLC1$^{TPR}$-JIP1$^{C-term}$ | 0.72 ± 0.02 | 3.19 ± 0.31 | −7.69 ± 0.24 | −0.32 | −7.37 |

DOI: https://doi.org/10.7554/eLife.38362.028

linked to DYT1, a neurological disorder that leads to uncontrollable muscular movements (*Ozelius et al., 1997*). In cells, TorsinA is predominantly localized in the lumen of the endoplasmic reticulum (ER) or nuclear envelope (NE) where its ATPase enzymatic activity requires the stimulation by one of two transmembrane cofactors, LAP1 (lamina associated polypeptide 1) or LULL1 (luminal domain like LAP1), via their luminal domains (*Zhao et al., 2013*). However, the exact biological function of TorsinA and other family members is at present uncertain although they clearly play a role in the integrity of the NE (*Laudermilch and Schlieker, 2016*). Interestingly, wild-type TorsinA expressed in brain CAD cells has been found co-localized with endogenous KLC1$^{TPR}$ at the distal end of processes, whereas mutant TorsinA$^{ΔGAG}$ remained confined to the cell body thus opening the possibility that the wild-type protein undergoes anterograde transport along MTs possibly acting as a molecular chaperone regulating kinesin-1 activity and/or cargo binding (*Kamm et al., 2004*). This intriguing function would imply the existence of an endogenous TorsinA cytosolic pool that might be available to interact with kinesin-1. Indeed, the literature suggests that oxidative or heat-shock stresses can induce TorsinA translocation from the ER to the cytosol (*Adam et al., 2017*; *Hewett et al., 2003*). Our structural study shows that the C-terminus of TorsinA binds, like JIP1, preferentially to KLC1$^{TPR}$ with a similar low micromolar dissociation constant. In addition to the interaction with the Y-acidic core, KLC1$^{TPR}$ appears also perfectly poised to stabilize the additional negatively charged residue at position $p^{+3}$ as well as the more divergent peptide region N-terminal of the Y-acidic core. These interactions define a rather robust molecular interface that, on the basis of structural considerations, warrants a deeper analysis of a possible functional role of TorsinA as a kinesin-1 adaptor/cargo. Interestingly, a comparison of the KLC1$^{TPR}$-TorsinA$^{C-term}$ structure with that of the recently solved complex between the Torsin$^{AAA+}$ domain and its LULL1 activator reveals that the TorsinA$^{C-term}$ region can undergo a dramatic structural transition. The TorsinA$^{C-term}$ segment that is part of the Torsin$^{AAA+}$ domain folds into a α-helix in the complex with LULL1 playing a role in its stabilization (*Demircioglu et al., 2016*). On the other hand, TorsinA$^{C-term}$ assumes a completely extended conformation when in complex with KLC1$^{TPR}$. This change in secondary structure is reminiscent of that undergone by the C-terminal domain (CTD) of the tubulin-like FtsZ upon binding to different regulatory proteins where the CTD folds into a α-helix when bound to the *E. coli* FtsZ-binding domain of ZipA and *Thermotoga maritima* FtsA (*Mosyak et al., 2000*; *Szwedziak et al., 2012*) while it takes an extended conformation when bound to SlmA from *E. coli*, *Vibrio cholera*, and *Klebsiella pneumonia* (*Schumacher and Zeng, 2016*). This plasticity has been suggested to be important for the versatile recognition of different regulators (*Schumacher and Zeng, 2016*). Similarly, it could be a mechanism utilized by TorsinA to select different partners, in this case, in alternative subcellular locations.

The role of cargo adaptors in kinesin-1 motor activation is a complex and an incompletely understood one. Remarkably, adaptors featuring W-acidic motifs appear to encode both motor recognition (via KLC$^{TPR}$ binding) and activation capacities. For example, an artificial transmembrane protein containing either of the two cytosolic W-acidic motifs of calsyntenin was shown to induce kinesin-1's vesicular association and anterograde transport in a KLC-dependent manner (*Kawano et al., 2012*). Moreover, these motifs can also functionally substitute for homologous regions within the vaccinia virus KLC binding protein A36 and activate transport (*Dodding et al., 2011*) or when fused to LAMP1, promote lysosome transport (*Pu et al., 2015*). Differently, while the C-terminal Y-acidic region of JIP1 is sufficient to recruit KLC1 to vesicles, it is not able to activate transport (*Kawano et al., 2012*). Indeed, JIP1 requires interactions with the coiled-coil region of KHC and the KHC-tail binding protein FEZ1, or JIP3, a separate member of the JIP family, for full activity (*Blasius et al., 2007*; *Fu and Holzbaur, 2014*; *Hammond et al., 2008*; *Satake et al., 2013*).

Recently, a clear cooperative role of JIP1 and JIP3 has been identified for the anterograde axonal transport of TrkB receptors (*Sun et al., 2017*), further supporting previous results that indicated that JIP1 and JIP3 are co-transported in neuronal cells (*Hammond et al., 2008*). The structural results presented here in conjunction with our previous studies allow us to formulate some hypotheses as to why Y-acidic and W-acidic motifs induce different functional responses. Our previous work has demonstrated that binding of W-acidic motifs displaces an intramolecular interaction between the KLC$^{TPR}$ and the LFP-acidic region (*Yip et al., 2016*). Moreover, binding of W-acidic motifs or disruption of the LFP-acidic intramolecular interaction triggers global changes in the KLC conformation, promotes kinesin-1 activity and results in an additional binding site on KHC-tail becoming accessible (*Sanger et al., 2017*; *Yip et al., 2016*). A key piece of data supporting this model is the observation that inclusion of the inhibitory LFP-containing sequence on KLC1/2$^{TPR}$ substantially reduces its affinity for W-acidic peptides. On the other hand, inclusion of the LFP-acidic sequence has very little effect on KLC1$^{TPR}$ affinity for the JIP1$^{C-term}$ Y-acidic peptide. This suggests that Y-acidic binding might co-exist with LFP-acidic binding and therefore cannot alone initiate this activation pathway. Two possibilities then present themselves – firstly, cooperative interactions with JIP3 serve to displace the LFP-acidic linker, driving the same activation pathway. Alternatively, the LFP-acidic linker remains bound and there is an as yet uncharacterized activation pathway triggered by JIP1/JIP3 binding that is more reliant on the additional direct adaptor interactions with KHC. Our ITC and mass spectrometry data suggest the possibility that could be driven by steric effects resulting from the JIP3$^{LZ2}$-mediated dimerization of KLC1$^{TPR}$ within the kinesin-1 tetramer, a notion further substantiated by the recent publication of a KLC2$^{TPR}$-JIP3$^{LZ2}$ structure that shows two TPR domains (in their 'open' conformation) symmetrically bound to the JIP3$^{LZ2}$ coiled-coil (*Cockburn et al., 2018*). However, our data strongly argue that a functional JIP1-JIP3 transport complex would have its TPR domains in their closed conformation induced by Y-acidic binding. Structural insight into ternary KLC1-JIP1-JIP3 complexes and the tetrameric kinesin-1 tail region will be required to fully address the question of co-operativity in kinesin-1-mediated transport.

## Materials and methods

### Cloning

Codon-optimized DNA sequences encoding the TPR domain of mouse KLC1 (residues 205 – 496, Uniprot Q5UE59, KLC1$^{TPR}$) fused either to the C-terminal 11 amino acids of human JIP1 (residues 701 – 711, Uniprot Q9UQF2, JIP1$^{C-term}$) or the C-terminal 12 amino acids of mouse TorsinA (residues 322 – 333, Uniprot Q9ER39, TorsinA$^{C-term}$) linked via a (Thr-Gly-Ser)$_{5 \text{ or } 10}$ flexible connector were purchased from Genscript and subcloned between the NdeI/XhoI sites of a pET28 vector (Novagen). This strategy allows for the expression of chimeric proteins that will be identified as KLC1$^{TPR}$-JIP1$^{C-term}$ and KLC1$^{TPR}$-TorsinA$^{C-term}$, respectively, bearing a thrombin-cleavable N-terminal hexa-histidine tag. Nanobodies were generated by the VIB Nanobody Service Facility of Vrije Universiteit Brussel (VUB) following the immunization of a llama with purified KLC1$^{TPR}$-JIP1$^{C-term}$ according to published protocols. Out of a panel of 36 antigen-specific nanobodies supplied in a pMECS vector, we subcloned a subset representing the most divergent amino acidic sequences between the PstI/BstEII sites of a pHEN6c vector. This allowed the expression of C-terminally His$_6$-tagged nanobodies in the periplasmic space of *E. coli*. The LZ2 region of mouse JIP3 corresponding to residues (416-485) was expressed as Prescission-cleavable GST-fusion. FLAG-tagged JIP1 and JIP3 for expression in mammalian cells were obtained from Addgene as clones 52123 and 53458, respectively.

### Protein expression and purification

Chimeric proteins were expressed in the *E.coli* BL21(DE3) strain. Briefly, single colonies were picked and grown at 21°C overnight. Small-scale overnight bacterial cultures were used to inoculate 8 × 0.5L cultures that were incubated at 37°C until they reached an OD600 of 0.4 – 0.6. The temperature was then lowered to 18°C and protein synthesis was induced by the addition of 500 µM isopropyl b-D-1-thiogalactopyranoside (IPTG) for 16 hr. Cells were harvested by centrifugation at 5000 g for 15 min at 4°C and resuspended in 50 mM 4-(2-hydroxyethyl)−1-piperazineethanesulfonic acid (HEPES) buffer at pH 7.5, 500 mM NaCl, 5 mM β-mercaptoethanol supplemented with protease inhibitor cocktail (Roche) and 5 U/ml of Benzonase endonuclease (Merck). Cell lysis was

accomplished by sonication. Insoluble material was sedimented by centrifugation at 16500 g for 1 hr at 4°C and the supernatant microfiltered using a 0.22 µm pore size filter prior to loading on a HisTrap column (GE Healthcare) pre-equilibrated with lysis buffer for immobilized metal affinity chromatography (IMAC). Proteins were eluted with an imidazole gradient and fractions containing the target protein collected and dialysed overnight at 4°C against imidazole-free lysis buffer. The purification tag was cleaved by incubating the dialysed sample for approximately 5 hr at room temperature in the presence of thrombin protease covalently bound to agarose beads (Thrombin CleanCleave Kit, Sigma). Beads were removed by filtration over a gravitational column and the eluate further microfiltered using 0.22 µm pore size filter prior to loading it on a HisTrap column (GE Healthcare) pre-equilibrated with lysis buffer. Untagged material present in the flow-through fraction was collected, concentrated, and further purified by size exclusion chromatography (SEC) on a 16/60 HiLoad Superdex 75 column (GE Healthcare) equilibrated with 25 mM HEPES, pH 7.5, 150 mM NaCl, and 5 mM β-mercaptoethanol.

Nanobodies were expressed in *E.coli* WK6 cells. Protein expression in TB medium was induced by addition of 1 mM IPTG at an OD600 of 0.9 – 1.0, and cells were grown overnight at 28°C. The periplasm fraction was harvested by osmotic shock, and Nanobodies were purified by IMAC and SEC using a HiLoad Superdex 75 column (GE Healthcare). The Nb:KLC1$^{TPR}$-JIP1$^{C-term}$ and Nb:KLC1$^{TPR}$-TorsinA$^{C-term}$ complexes used for crystallization were obtained by mixing the individual components with a 1:1 molar ratio and allowing incubation on ice for 30 min. The complexes were further purified by SEC on a 16/60 HiLoad Superdex 75 column (GE Healthcare).

## Crystallization

All crystallization experiments were performed using the vapor diffusion setup at 18°C and a 1:1 protein:precipitant ratio in 400 nl sitting drops dispensed with the aid of Mosquito crystallization robot (TTP LabTech). Crystals of the Nb:KLC1$^{TPR}$-JIP1$^{C-term}$ complex grew from a protein solution concentrated at ~10 mg/ml in the presence of a non-buffered reservoir solution containing 0.04 M potassium phosphate monobasic, 16% polyethylene glycol (PEG) 8000 (w/v) and 20% glycerol (v/v). For Nb:KLC1$^{TPR}$-TorsinA$^{C-term}$, the complex was also concentrated at ~10 mg/ml and crystals were obtained in the presence of 0.1 M HEPES pH 7.5, 10% propan-2-ol (v/v), 20% PEG 4000 (w/v). For cryoprotection, the latter crystals were briefly transferred to a reservoir solution in which the propan-2-ol concentration was increased to 30% (v/v).

## X-ray data collection, processing and structure solution

Datasets at the 2.7 Å and 2.3 Å resolution were measured at the P14 beam line of Petra III (Hamburg, Germany) and at the I04 beam line of Diamond Light Source (Didcot, United Kingdom) for Nb:KLC1$^{TPR}$-JIP1$^{C-term}$ and Nb:KLC1$^{TPR}$-TorsinA$^{C-term}$, respectively. Although the growth conditions are different both complexes crystallize in the monoclinic space group *C*2 and display similar cell dimensions (*Table 1*). Data were processed using the *xia*2 pipeline (*Kabsch, 2010*; *Waterman et al., 2016*; *Winter et al., 2013*) and the structures were solved by the molecular replacement (MR) technique using the software package *Phaser* (*McCoy et al., 2007*). Based on our previous experience with W-acidic binding (*Pernigo et al., 2013*), we anticipated a TPR domain closure upon Y-acidic peptide binding. Therefore, we carried out molecular replacement trials with the C-terminal portion of KLC1 bound to the Nb that we had solved previously at low resolution. Model building and crystallographic refinement was performed using *COOT* (*Emsley and Cowtan, 2004*) and *BUSTER* (*Bricogne et al., 2017*), respectively. A summary of data collection and refinement statistics are shown in *Table 1*. Structural images were prepared with PyMol (Schrödinger).

## Isothermal titration calorimetry (ITC)

Samples for ITC measurements were extensively dialyzed in buffer composed of 50 mM HEPES, pH 7.5, 500 mM NaCl, and 5 mM 2-mercaptoethanol. ITC experiments were conducted on a MicroCal ITC (MicroCal Inc.) instrument at a temperature of 20°C. GST-JIP3$^{LZ2}$ (550 µM) was loaded in the syringe and used as titrant while either His$_6$-KLC1$^{extTPR}$, His$_6$-KLC1$^{TPR}$-JIP1$^{C-term}$ or His$_6$-KLC1$^{extTPR}$-JIP1$^{C-term}$ were loaded in the cell and employed as analytes at the final concentration of 25 µM. Data were corrected for heats of dilution of the protein solution. Binding constants and other

thermodynamic parameters were calculated by fitting the integrated titration data assuming a single set of binding sites using the Origin software package (OriginLab).

### Fluorescence polarization (FP)

N-terminal TAMRA-conjugated peptides and non-conjugated peptides used for FP and competition measurements were supplied either by Bio-Synthesis (Lewisville, TX) or by Pepceuticals Ltd. (Enderby Leicestershire, UK). Sequences were as follows: JIP1[C-term], YTCPTEDIYLE; TorsinA[C-term] TVFTKLDYYLDD; SKIP[WD], STNLEWDDSAI. JIP1[amidated C-term] has the same sequence as JIP1[C-term] but features an amidated C-terminus. Measurements were performed on a BMG Labtech PolarStar Omega plate reader at 20°C by incubating 300 nM TAMRA-labeled peptides with the indicated protein at increasing concentrations in 25 mM HEPES, pH 7.5, 5 mM 2-mercaptoethanol, and NaCl in the concentration range 85 – 500 mM. Estimation of equilibrium dissociation constant ($K_D$) values was performed assuming a one-site specific-binding model using the Prism package (GraphPad Software). For competition experiments, a mixture of TAMRA-JIP1[C-term] and KLC1[TPR] at 300 nM and 2 µM, respectively, was incubated with increasing concentrations of unlabeled SKIP[WD] peptide in 25 mM HEPES, pH 7.5, 150 mM NaCl, 5 mM 2-mercaptoethanol buffer supplemented by 5% (v/v) DMSO. The concentration-dependent decrease in FP signal was fitted to a sigmoidal equation to derive IC50. $K_I$ estimation was performed using the online tool (http://sw16.im.med.umich.edu/software/calc_ki/online tool) based on *Nikolovska-Coleska et al., 2004*). Data points are the mean of three replicates.

### Native mass spectrometry

The protein complex (25 µl) was buffer exchanged into 150 mM ammonium acetate, pH 7.5 using Micro Bio-Spin six columns (Bio-Rad, CA). The buffer exchanged sample was infused into the mass spectrometer, a Synapt G2Si qTOF (Waters, MA), using a 20:80 Au:Pd coated borosilicate capillary into a nanoflow electrospray source. The instrument was calibrated using a caesium iodide mixture. The mass spectrometer was run in positive mode with a capillary voltage of 1.2 – 2.0 kV, a sampling cone of 40 – 80 V and a source temperature of 45°C. Mass spectra were processed using MassLynx V4.1 (Waters, MA).

### Immunoprecipitation

HeLa cells were maintained in DMEM supplemented with 10% FBS, L-glutamine and penicillin/streptomycin, and cultured in a humidified 5% $CO_2$ environment. For immunoprecipitation experiments, $1 \cdot 10^6$ cells were seeded on 10 cm dishes and transfected the next day with vectors expressing FLAG-JIP1 and HA-KLC1[TPR]. KLC1[TPR] variants Q222A, R266S, V305T, N343S, N386S, N469S, E484A, and E488A were produced by site-directed mutagenesis using the Quickchange kit and tested alongside wild-type HA-KLC1[TPR]. After 24 hr, transfected cells were lysed in 1 ml of lysis buffer (25 mM HEPES pH 7.5, 150 mM NaCl, 0.1% Nonidet P-40, 0.1% Triton X-100 containing a protease inhibitor mixture (Roche)) for 10 min before centrifugation at 13000·g for 10 min at 4°C. Supernatants were added to 40 µl of anti-FLAG M2 agarose (A2220, Sigma) and incubated at 4°C for 2 hr on a rotator. After binding, beads were washed four times in lysis buffer, resuspended in 100 µl of buffer and 25 µl of SDS-loading buffer before boiling. 20 µl of samples were subjected to SDS-PAGE and analyzed by western blot using antibodies against FLAG and HA. FLAG-JIP1/FLAG-JIP3 experiments were performed under the same conditions, using anti-HA agarose (A2095, Sigma)

## Acknowledgements

The X-ray crystallography work was conducted at beamline I04 of Diamond Light Source (Didcot, UK) and at beamline P14 of Petra III/DESY (Hamburg, Germany). We are grateful to the staff scientists working at these beamlines for their excellent technical support.

# Additional information

## Funding

| Funder | Grant reference number | Author |
| --- | --- | --- |
| Biotechnology and Biological Sciences Research Council | BB/L006774/1 | Mark P Dodding Roberto A Steiner |
| Biotechnology and Biological Sciences Research Council | BB/S000917/1 | Mark P Dodding |
| Biotechnology and Biological Sciences Research Council | BB/S000828/1 | Roberto A Steiner |

The funders had no role in study design, data collection and interpretation, or the decision to submit the work for publication.

## Author contributions

Stefano Pernigo, Magda S Chegkazi, Yan Y Yip, Formal analysis, Validation, Investigation, Writing—review and editing; Conor Treacy, Giulia Glorani, Soi Bui, Investigation, Writing—review and editing; Kjetil Hansen, Argyris Politis, Validation, Investigation, Writing—review and editing; Mark P Dodding, Roberto A Steiner, Conceptualization, Resources, Data curation, Formal analysis, Supervision, Funding acquisition, Validation, Investigation, Visualization, Methodology, Writing—original draft, Project administration, Writing—review and editing

## Author ORCIDs

Magda S Chegkazi (iD) http://orcid.org/0000-0002-0855-2681
Kjetil Hansen (iD) https://orcid.org/0000-0002-0085-8440
Argyris Politis (iD) http://orcid.org/0000-0002-6658-3224
Mark P Dodding (iD) http://orcid.org/0000-0001-8091-6534
Roberto A Steiner (iD) http://orcid.org/0000-0001-7084-9745

## Decision letter and Author response

Decision letter https://doi.org/10.7554/eLife.38362.035
Author response https://doi.org/10.7554/eLife.38362.036

# Additional files

## Supplementary files

• Transparent reporting form
DOI: https://doi.org/10.7554/eLife.38362.029

## Data availability

Diffraction data and coordinates are publicly available in PDB under the accession codes 6FUZ and 6FV0

The following datasets were generated:

| Author(s) | Year | Dataset title | Dataset URL | Database and Identifier |
| --- | --- | --- | --- | --- |
| Pernigo S, Dodding MP, Steiner RA | 2018 | Crystal structure of the TPR domain of KLC1 in complex with the C-terminal peptide of JIP1 | https://www.rcsb.org/structure/6FUZ | RCSB Protein Data Bank, 6FUZ |
| Pernigo S, Dodding MP, Steiner RA | 2018 | Crystal structure of the TPR domain of KLC1 in complex with the C-terminal peptide of TorsinA | https://www.rcsb.org/structure/6FV0 | RCSB Protein Data Bank, 6FV0 |

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
