## [Decision Letter]

[Editors’ note: this article was originally rejected after discussions between the reviewers, but the authors were invited to resubmit after an appeal against the decision.]

Thank you for submitting your work entitled "Structural basis for isoform-specific kinesin-1 recognition of Y-acidic cargo adaptors" for consideration by *eLife*. Your article has been reviewed by three peer reviewers, one of whom is a member of our Board of Reviewing Editors, and the evaluation has been overseen by a Senior Editor. The reviewers have opted to remain anonymous.

Our decision has been reached after consultation between the reviewers. Based on these discussions and the individual reviews below, we regret to inform you that your work will not be considered further for publication in *eLife*.

While all of the reviewers appreciated that this work was well executed, after a discussion amongst the reviewers there was a clear consensus that the work did not provide a significant enough advance over other literature in the field to merit publication in *eLife*. The reviewers hope their feedback will be helpful for revising for publication in another journal.

Reviewer #1:

How kinesin motors attach to their cargos remains relatively poorly understood. Kinesin-1 family members (Kif5A-C) exist as heterotetramers with kinesin light chains (KLC1-4). In this study, Pernigo et al. identify a recognition motif that is important for the interaction between KLC1 and a cargo adaptor (JIP1) and a candidate cargo (TorsinA). The authors describe a Y-acidic motif that, as opposed to the previously identified non-KLC isoform specific W-acidic motif, has a strong preference for KLC1 binding. The Y and W-acidic motifs are partially overlapping and adaptors that bind to these sites compete for interacting with the KLC1. This work is technically well executed and provides new information for how KLCs interact with adaptors or cargos. However, given previous work in the field, including studies from this group, I do not find the results presented here to be a significant enough advance to merit publishing in *eLife*. Journals that focus more on cell biology and the cytoskeleton would be a more appropriate venue for this work.

1) The argument suggesting that KLC1 specificity over KLC2 is due to the N to S substitutions at residues 343 and 469 is not fully convincing given the limited cellular validation. The authors mutate N343S and N469S in KLC1. Do the converse substitutions in KLC2 increase the Y-acidic motif binding affinity for KLC2? It would also be beneficial to characterize the change in binding affinities for the selected mutants as performed for the WT KCL1^TPR^ and JIP1 peptide in Figure 1D.

2) The proposed model for how JIP1 and JIP3 cooperate to activate KLC1 is quite speculative. The data in which JIP1 peptide binding to the KLC1 TPR domain does not affect JIP3 binding affinity is not fully supportive of a cooperative function. The proposed dimerization mediated by KLC1 TPR and JIP3^LZ2^ should also be further explored. The use of larger protein domains or full-length proteins might be beneficial to account for contributions from other regions of these proteins.

*Reviewer #2:*

Pernigo, Dodding, Steiner and colleagues have identified a novel interaction motif, which they call "Y-acidic," in JIP1 and TorsinA that enables binding specifically to KLC1 but not KLC2. Intriguingly, the Y-acidic motif binds to KLC1 in such a way that it does not interfere with the binding of peptides that affect KLC autoinhibition. This suggests that adaptors with a Y-acidic motif would not automatically trigger kinesin-1 activation, which is in contrast to adaptors with a W-acidic motif. Their Y-acidic-adaptor model is thought-provoking and appealing when considering mechanisms that coordinate cargo binding and motor activation. Overall, the manuscript is well written, and the data are clearly presented.

Some questions arise as the authors attempt to translate their structural observations into how the Y-acidic motif would mediate cellular functions (Figure 3D). In their IP experiments, to show that these KLC mutations are acting specifically, the authors should include a mutation that is not predicted to interfere with JIP1 binding. It would also be nice if they could introduce reciprocal mutations in JIP1 to directly test the Y-acidic motif. Similarly, can the authors mutate the KLC2 serines equivalent to N343 and N469 in KLC1 and cause KLC2 to bind JIP1? It would be nice if the authors could provide evidence via cell imaging that co-localization of KLC1 and JIP1 in cells is regulated by the Y-acidic motif, or at least that a cellular function is regulated by the Y-acidic motif mediating an interaction with KLC1/kinesin-1. As it stands, it is not clear whether this interaction may ultimately be redundant with other points of interaction between JIP1 and KLC/kinesin-1 in cells.

Reviewer #3:

Pernigo et al. report the binding mode of adaptor proteins to the light chain of kinesin-1 (KLC). They have in particular determined the structure of the binding motif of Jip1 and of Torsin1, C-terminal peptides of about 10 residues, to the TPR domain of KLC1, allowing them to propose a mechanism for the specificity of these adaptors for KLC1 vs. KLC2. To crystallize these complexes, they used a fusion KLC1-peptide protein in addition to nanobodies as crystallization chaperones, an approach that is worth mentioning. And more generally, the structural results justify publishing in *eLife*. I have nevertheless several concerns which are listed below.

1) The dissociation constant of the complex of KLC1-TPR with the C-term peptide of Jip1 or of TorsinA is in the μM range. Yet the peptides interact quite extensively with KLC1 and one could have expected a higher affinity. The authors mention that the measured affinities are consistent with the results of Zhu et al. But can we exclude an interference of the TAMRA probe used in the fluorescence experiments? An orthogonal method, preferably using unlabeled peptides (e.g. by ITC), performed in the same lab with the exact same buffer and proteins, would ascertain the affinity values.

I note that many interactions are of electrostatic nature while the buffer used contains 150 mM NaCl. It is therefore possible that the ionic strength has a substantial effect on the affinity.

2) According to Figure 2B, the TPR binding surface is mostly basic. This is puzzling because the Jip1 C-term peptide contains several hydrophobic residues (in particular the IYL motif). At which level is the electrostatic potential depicted? The same remark applies also to Figure 3—figure supplement 2A and to the TorsinA peptide.

3) Figure 3C indicates that the C-term carboxyl group of Jip1 makes a salt bridge with KLC1. This point is rapidly mentioned in the text (subsection “KLC1^TPR^-Y-acidic peptide interface”, last paragraph) but not discussed further. It is striking that in both Jip1 and Torsin1 the interacting peptide includes the C-terminal of the protein, as is also the case of the PipB2 protein mentioned in the Discussion. This suggests that the C-term carboxyl contributes substantially to KLC binding. This point could be discussed further (and could be tested experimentally, using a carboxy C-term modified peptide in affinity experiments).

4) I am not an expert in ITC and I could be wrong, but the titration curve in Figure 5A seems weird to me and my understanding is that the first part of this titration curve was not used in the fitting curve. Is the experiment reproducible? For the least, an explanation is needed.

5) It is not clear what the goal of the MS experiment was and what could be concluded (Figure 5C). Do the authors suggest that TPR dimerization is mediated by JIP3^LZ2^ dimer (according to the title of Figure 5)? The MS data are not consistent with this view, because a KLC1-Jip1 dimer is detected in the absence of JIP3^LZ2^. In addition, the authors propose that the N-term helix of the TPR domain may mediate KLC1 dimerization. An alternative hypothesis would be that the TPR-Jip1 construct dimerizes via the Jip1 peptide of one molecule that would interact 'in trans' with the TPR domain of another molecule. A control with a construct comprising the TPR domain alone would be welcome. Moreover, because the experiment is performed with an excess of TPR-Jip1 over JIP3^LZ2^, a 2-1 stoichiometry is not unexpected. It seems to me that a conclusion can hardly be drawn from these data.

---

## [Author Response]

[Editors’ note: the author responses to the first round of peer review follow.]

We thank the reviewers and editors for the opportunity to submit a revised version of our manuscript by Pernigo et al. entitled 'Structural basis for isoform-specific kinesin1 recognition of Y-acidic cargo adaptors'. We believe that we have satisfactorily answered all points raised by the reviewers. We therefore hope that our revised manuscript will be accepted for publication in *eLife*.

We should also to point out that during the submission/revision process of our manuscript two papers relevant to this study have been published: Nguyen et al. JBC, 2018, 10.1074/jbc.RA118.003916 and Cockburn et al., 2018.

Nguyen et al. is a biochemical study (almost entirely ITC analyses) of the KLC1^TPR^JIP1^C-term^ interface. Their study is entirely consistent with our X-ray structure. In fact, the authors use the structural coordinates that we made publicly available at the preprint stage to discuss their results.

Cockburn et al. presents the X-ray crystal structure of KLC2^TPR^ in complex with the coiled-coil JIP3^LZ2^ region. This interaction was also explored by us using ITC and mass spectrometry and the results presented in the current manuscript. Our conclusions are supported by their structural results.

This high activity in the field by various laboratories highlights the timeliness of our work that we hope will be made available to the general readership of *eLife*.

Our detailed responses to the reviewers' comments are below.

Reviewer #1:How kinesin motors attach to their cargos remains relatively poorly understood. Kinesin-1 family members (Kif5A-C) exist as heterotetramers with kinesin light chains (KLC1-4). In this study, Pernigo et al. identify a recognition motif that is important for the interaction between KLC1 and a cargo adaptor (JIP1) and a candidate cargo (TorsinA). The authors describe a Y-acidic motif that, as opposed to the previously identified non-KLC isoform specific W-acidic motif, has a strong preference for KLC1 binding. The Y and W-acidic motifs are partially overlapping and adaptors that bind to these sites compete for interacting with the KLC1. This work is technically well executed and provides new information for how KLCs interact with adaptors or cargos. However, given previous work in the field, including studies from this group, I do not find the results presented here to be a significant enough advance to merit publishing in eLife. Journals that focus more on cell biology and the cytoskeleton would be a more appropriate venue for this work.1) The argument suggesting that KLC1 specificity over KLC2 is due to the N to S substitutions at residues 343 and 469 is not fully convincing given the limited cellular validation. The authors mutate N343S and N469S in KLC1. Do the converse substitutions in KLC2 increase the Y-acidic motif binding affinity for KLC2? It would also be beneficial to characterize the change in binding affinities for the selected mutants as performed for the WT KCL1^TPR^ and JIP1 peptide in Figure 1D.

N343 and N469 in KLC1 correspond to S328 and S454 in KLC2, respectively. As requested, we have determined *K*_D_ values by FP for single (KLC2^TPR S328N^ and KLC2^TPR S454N^) as well as double (KLC2^TPR S328,454N^) KLC2 variants using both JIP1^Cterm^ and TorsinA^C-term^ peptides. As suggested by the structure the KLC2^TPR S328,454N^ variant binds both C-terminal peptides with an affinity essentially identical to that of KLC1^TPR^. These results are now presented in an additional panel in Figure 3 (panel F) and in a novel Figure 3—figure supplement 5 for JIP1^C-term^ and TorsinA^C-term^, respectively. This experiment therefore further strengthens the argument made in our original submission.

2) The proposed model for how JIP1 and JIP3 cooperate to activate KLC1 is quite speculative. The data in which JIP1 peptide binding to the KLC1 TPR domain does not affect JIP3 binding affinity is not fully supportive of a cooperative function. The proposed dimerization mediated by KLC1 TPR and JIP3^LZ2^ should also be further explored. The use of larger protein domains or full-length proteins might be beneficial to account for contributions from other regions of these proteins.

Various authors have shown that JIP1^C-term^ binding to KLC1^TPR^ is necessary but not sufficient for kinesin-1 mediated transport of JIP1. Additionally, multiple studies have indicated JIP3 as a protein that co-operates with JIP1 to promote transport by kinesin-1 (Sun et al., 2017, Satake et al., 2013, Hammond et al., 2018).

To better investigate this effect in the context of the present study, we have added new immunoprecipitation data in Figure 5A using full length JIP1 and JIP3 as requested. These data demonstrate that co-operative effect and importantly, show its dependence on the Y-acidic interface as defined by the R266S mutation.

Our ITC experiments that looked at JIP3^LZ2^ binding to JIP1^C-term^-loaded and JIP1^C-term^free KLC1^TPR^ (in the revised version also in the presence of the LFP-acidic region as requested by another reviewer) indicate that i) neither JIP1^C-term^ peptide loading nor ii) the presence of the auto-inhibitory 'LFP-acidic' KLC1^TPR^ region has an effect on JIP3^LZ2^ affinity, which as the reviewer indicates, point to contributions from other regions of these proteins.

Taken together, we conclude that JIP1 and JIP3 do act co-operatively, via KLC1^TPR^, to recruit kinesin-1 and that this requires KLC1^TPR^ binding to JIP1^C-term^. However, key interactions mediating this effect lie outside of the direct KLC^TPR^ domain interfaces. This point is clarified in the final Discussion paragraph.

Reviewer #2:[…] Some questions arise as the authors attempt to translate their structural observations into how the Y-acidic motif would mediate cellular functions (Figure 3D). In their IP experiments, to show that these KLC mutations are acting specifically, the authors should include a mutation that is not predicted to interfere with JIP1 binding. It would also be nice if they could introduce reciprocal mutations in JIP1 to directly test the Y-acidic motif.

We have now included new data presented in Figure 3—figure supplement 4A that shows that three mutations (Q222A, E484A, E488A) that lie just outside of the Y-acidic interface (see model inFigure 3—figure supplement 4B) do not interfere with binding, in contrast to R266S that is provided for comparison on the same blot. In addition, a new reciprocal IP (IP for HA rather than FLAG) for R266S is presented in Figure 5A that further reinforces the specificity of assays used here. Finally, in new FP experiments, amidation of the JIP1^C-term^ peptide to remove the carboxy group on the C-terminus eliminates the interaction (Figure 3—figure supplement 1) It is also worth highlighting here the recent study by Nguyen et al. (JBC 2018) that incorporates a variety of other mutations in JIP1^C-term^ that similarly inhibit binding to KLC1^TPR^.

Similarly, can the authors mutate the KLC2 serines equivalent to N343 and N469 in KLC1 and cause KLC2 to bind JIP1?

This point was also raised by reviewer 1.This has been done and we have determined *K*_D_ values by FP for single (KLC2^TPR S328N^ and KLC2^TPR S454N^) as well as double (KLC2^TPR S328,454N^) KLC2 variants using both JIP1^C-term^ and TorsinA^C-term^ peptides. As suggested by the structure, the KLC2^TPR S328,454N^ variant binds both C-terminal peptides with an affinity essentially identical to that of KLC1^TPR^. These results are now presented in an additional panel in Figure 3 (panel F) and in a novel Figure 3—figure supplement 5 for JIP1^C-term^ and TorsinA^C-term^, respectively.

It would be nice if the authors could provide evidence via cell imaging that co-localization of KLC1 and JIP1 in cells is regulated by the Y-acidic motif, or at least that a cellular function is regulated by the Y-acidic motif mediating an interaction with KLC1/kinesin-1. As it stands, it is not clear whether this interaction may ultimately be redundant with other points of interaction between JIP1 and KLC/kinesin-1 in cells.

The need for an intact JIP1 C-terminus (which harbours the 'Y-acidic' motif) for kinesin-1 dependent JIP1 transport is very well documented in the literature. For example, in the first paper describing the JIP1-KLC1 interaction, the crucial tyrosine of the Y-acidic motif was shown to be required for the kinesin-1 dependent accumulation of JIP1 in neurite tips (Verhey et al., 2001, Figure 4-5). This was further substantiated in a later paper from the Verhey lab (Hammond et al., 2008) which examined co-operative transport of JIP1 and JIP3 and confirmed that the C-terminal/Y residue was required for efficient transport of the JIP1-JIP3 complex (Figure 7). Kawano et al., 2012 also confirmed that the C-terminal 11 amino acids of JIP1 were sufficient to recruit KLC1 to vesicles although not to activate kinesin dependent transport (Figure 7). Additionally, Satake et al., 2013 show by coIP and IF (Figure 1) that while full-length GFP-JIP1 accumulates at neurite tips in a kinesin-1 dependent manner, GFP-JIP1-dCT (dCT = delta C-terminus, lacking the final four amino acids that are an integral part of the Y-acidic motif) does not. GFPJIP1-dCT shows fluorescence intensity at the tips comparable to that of GFP alone. In this paper the GFP-JIP1-dCT construct is even used as a 'de facto' negative control to assess the relative impact of other JIP1 mutations on kinesin-1 recruitment. Although all the above references were already citied in the original submission we have further clarified in the revised text that an intact JIP1 C-terminus (containing the Y-acidic motif) is necessary (although not sufficient) for kinesin-1 transport mediated by KLC1^TPR^ recruitment.

Experimentally, our new IP data presented in Figure 5A using full length JIP1 and JIP3 demonstrates that they interact co-operatively with KLC1^TPR^ and, importantly, show its dependence on the Y-acidic interface as defined by the R266S mutation, further highlighting the importance of this interaction site. The discussion has been expanded to include further consideration of this point.

Reviewer #3:Pernigo et al. report the binding mode of adaptor proteins to the light chain of kinesin-1 (KLC). They have in particular determined the structure of the binding motif of Jip1 and of Torsin1, C-terminal peptides of about 10 residues, to the TPR domain of KLC1, allowing them to propose a mechanism for the specificity of these adaptors for KLC1 vs. KLC2. To crystallize these complexes, they used a fusion KLC1-peptide protein in addition to nanobodies as crystallization chaperones, an approach that is worth mentioning. And more generally, the structural results justify publishing in eLife. I have nevertheless several concerns which are listed below.1) The dissociation constant of the complex of KLC1-TPR with the C-term peptide of Jip1 or of TorsinA is in the μM range. Yet the peptides interact quite extensively with KLC1 and one could have expected a higher affinity. The authors mention that the measured affinities are consistent with the results of Zhu et al. But can we exclude an interference of the TAMRA probe used in the fluorescence experiments? An orthogonal method, preferably using unlabeled peptides (e.g. by ITC), performed in the same lab with the exact same buffer and proteins, would ascertain the affinity values.

At the request of this reviewer we have determined thermodynamic parameters for the binding of unlabelled JIP1^C-term^ to KLC1^TPR^ using ITC. The experiment was carried out at 500mM NaCl as KLC1^TPR^ tends to precipitate in the cell at low salt concentration upon stirring. FP measurements at this NaCl concentration give *K*_D_ = 10.37 ± 0.45 µM and ITC measurements give *K*_D_ = 11.66 ± 1.85 µM. The two measurements are therefore entirely consistent with each other and confirm that the TAMRA label does not interfere with binding. The ITC data and relative thermodynamic parameters for this experiment are reported in a novel Figure 1—figure supplement 2 in the revised manuscript.

I note that many interactions are of electrostatic nature while the buffer used contains 150 mM NaCl. It is therefore possible that the ionic strength has a substantial effect on the affinity.

The interaction between the KLC1^TPR^ domain and Y-acidic peptides elucidated in this work and also that between KLC2^TPR^ and W-acidic peptides studied in our previous work (Pernigo et al., 2013) has an electrostatic component (peptides are acidic and the concave surfaces of KLC^TPR^ domains are positively charged). The effect of ionic strength of the interaction between KLC2^TPR^ with and W-acidic peptides of the cargo adaptor SKIP was investigated to some extent in our 2013 paper. At the request of this reviewer we have now also explored the effect of varying NaCl concentrations for both the KCL1^TPR^-JIP1^C-term^ and KLC1^TPR^-TorsinA^C-term^ interactions. In the case of KCL1^TPR^-JIP1^C-term^, *K*_D_ values are 0.48 ± 0.06 µM and 10.37 ± 0.45 µM at [NaCl] = 85 and 500 mM, respectively, while for KCL1^TPR^TAMRA-TorsinA^C-term^*K*_D_ values are 1.23 ± 0.10 µM and 29.9 ± 2.8 µM at the same NaCl concentrations, respectively. Therefore, as expected from our previous work, there is an electrostatic component to the binding. Our choice of [NaCl]=150mM for most FP measurements allows for high affinity interactions while minimising KLC1^TPR^ precipitation. All FP measurements at varying NaCl concentrations are now presented in a novel Figure 1—figure supplement 1 in the revised manuscript.

2) According to Figure 2B, the TPR binding surface is mostly basic. This is puzzling because the Jip1 C-term peptide contains several hydrophobic residues (in particular the IYL motif). At which level is the electrostatic potential depicted? The same remark applies also to Figure 3—figure supplement 2A and to the TorsinA peptide.

We used consistent threshold values (+10*k*_B_T/e (blue) to -10*k*_B_T/e (red)) to depict the electrostatic potential in all surface representations. As mentioned in our reply to the previous point and consistent with this reviewer's suggestion the binding does contain an electrostatic component. Therefore we consider not too puzzling that the overall character of the receptor surface is basic given the overall acidic nature of the cargo adaptor peptides. Threshold values are now explicitly mentioned in the relevant figure legends.

3) Figure 3C indicates that the C-term carboxyl group of Jip1 makes a salt bridge with KLC1. This point is rapidly mentioned in the text (subsection “KLC1^TPR^-Y-acidic peptide interface”, last paragraph) but not discussed further. It is striking that in both Jip1 and Torsin1 the interacting peptide includes the C-terminal of the protein, as is also the case of the PipB2 protein mentioned in the Discussion. This suggests that the C-term carboxyl contributes substantially to KLC binding. This point could be discussed further (and could be tested experimentally, using a carboxy C-term modified peptide in affinity experiments).

Both Jip1 and TorsinA use their carboxy termini in the interaction with KLC1^TPR^.

The suggestion by this reviewer to use a C-term modified peptide is an excellent one. To this end we have synthesized a modified TAMRA-JIP1^C-term^ featuring C-terminal amidation. We find that a free carboxylic group in JIP1 is absolutely required for the interaction. This piece of data is now presented in a novel Figure 3—figure supplement 1 in the revised manuscript.

4) I am not an expert in ITC and I could be wrong, but the titration curve in Figure 5A seems weird to me and my understanding is that the first part of this titration curve was not used in the fitting curve. Is the experiment reproducible? For the least, an explanation is needed.

The first part of the titration shown in Figure 5A (Figure 5D in the revised figure) was also used for fitting and it is not a question of reproducibility. The reason why the height of the first portion of this thermogram is different is because we used a smaller injection volume compared to subsequent ones in an attempt to better sample the initial portion of the curve. We have now mentioned this explicitly in the legend.

5) It is not clear what the goal of the MS experiment was and what could be concluded (Figure 5C). Do the authors suggest that TPR dimerization is mediated by JIP3^LZ2^ dimer (according to the title of Figure 5)? The MS data are not consistent with this view, because a KLC1-Jip1 dimer is detected in the absence of JIP3^LZ2^. In addition, the authors propose that the N-term helix of the TPR domain may mediate KLC1 dimerization. An alternative hypothesis would be that the TPR-Jip1 construct dimerizes via the Jip1 peptide of one molecule that would interact 'in trans' with the TPR domain of another molecule. A control with a construct comprising the TPR domain alone would be welcome. Moreover, because the experiment is performed with an excess of TPR-Jip1 over JIP3^LZ2^, a 2-1 stoichiometry is not unexpected. It seems to me that a conclusion can hardly be drawn from these data.

The main purpose of the experiment was to provide an orthogonal observation of the stoichiometry of two KLC1^TPR^ bound to the JIP3^LZ2^ coiled-coil dimer suggested by the ITC experiment. The MS data indeed show a complex of this stoichiometry. While our manuscript was under revision Cockburn et al. have published the crystal structure of the KLC2^TPR^-JIP3^LZ2^ complex that is entirely consistent with our observations. Given the availability of this new crystal structure we have moved our MS data to the supplementary information (Figure 5—figure supplement 1) and added the relevant reference to the text.